# PRISM: A Principled Framework for Supervised Disentanglement via Bipartite Factorization

## Abstract

Learning structured representations that partition information based on its semantic contents remains a central challenge in deep generative modeling. In light of the established theoretical impossibility of purely unsupervised disentanglement, we address the pragmatic and well-posed objective of bipartite factorization: separating the single factor of variation corresponding to the supervisory label from all other residual sources of variation. We introduce a principled framework that achieves this separation through a learning mechanism that routes most of the intra-class variation into a class-agnostic latent subspace. The design of this mechanism is guided by a formal, information-theoretic analysis, which provides quantitative bounds on the learning outcome. We conduct a series of targeted experiments designed to validate the proposed mechanism, demonstrating its ability to produce a factorized representation with quantifiably low leakage of supervised information into the residual subspace, and illustrating the effectiveness of the resulting factorization on downstream tasks requiring precise latent control, such as targeted attribute swapping and manipulation of stylistic features.

## 1 Introduction

Deep generative models that learn structured latent representations are foundational to achieving controllable synthesis and robust downstream task performance. An effective representation should isolate the distinct, explanatory factors of variation in the data, a property often referred to as disentanglement. While this goal has motivated extensive research, a significant theoretical result has revealed that the unsupervised learning of such representations is fundamentally ill-posed. Without inductive biases or some form of supervision, any dataset admits an infinite set of equally likely factorizations, making the recovery of a ground-truth disentangled representation unreliable (Locatello et al., 2019).

This impossibility result motivates a shift towards more constrained, yet highly practical, problem formulations. We address the well-posed objective of bipartite factorization: partitioning a latent space to isolate a single, supervised attribute from all other residual sources of variation. Such a representation is valuable for a range of applications, from targeted attribute editing to improving algorithmic fairness by isolating a sensitive attribute from other predictive features. Supervised methods typically realize the attribute–residual split via adversarial or split-encoder designs (Lample et al., 2017; Hadad et al., 2018; Zheng & Sun, 2019; Ding et al., 2020) and are primarily validated empirically; formal leakage guarantees between subspaces are uncommon.

Our core contribution is the Prototype-Regularized Information-Splitting Model (PRISM), a principled framework that achieves bipartite factorization of a latent space by combining adversarial learning with information-theoretic constraints. We identify a key mechanism that emerges from this synthesis: the framework orchestrates two competing processes, where an adversarial objective purges supervised information from one latent subspace while a structural bottleneck constrains the other. The key insight is that this deliberate orchestration creates a conflict that the optimization must resolve. By rendering the supervised subspace an inefficient channel for reconstruction details, the model is compelled to route this necessary information through the now class-agnostic residual subspace. This achieves a robust factorization as an emergent property of the system's design.

Our contributions are threefold. First, we introduce a constructive method that achieves bipartite factorization by imposing a tight information bottleneck on the supervised latent subspace. This constraint, in conjunction with a reconstruction objective, compels the model to route the necessary intra-class variation into a complementary residual subspace. Second, we provide a formal information-theoretic analysis of this process, grounded in the Information Bottleneck, which yields quantitative bounds that characterize both the suppression of supervised information from the residual subspace and the consequent displacement of unsupervised variation into it. Finally, we present a series of targeted experiments that validate the predictions of our formal analysis, showing that manipulating the identified architectural levers produces the expected changes in the information partition and demonstrating the utility of the resulting representation for downstream tasks such as targeted attribute swapping.

## 2 RELATED WORK

A central goal in representation learning is to discover and isolate the distinct factors of variation that govern the data. This pursuit has taken many forms, with early and influential approaches developing powerful mechanisms within supervised learning frameworks, often to enforce robustness and invariance. In domain adaptation, for example, adversarial training was used to learn features that are invariant to distribution shifts, improving cross-domain generalization (Ganin et al., 2016). This idea of adversarial invariance was carried into generative modeling - e.g., Fader Networks adversarially remove a designated semantic attribute from the latent code to enable controlled editing (Lample et al., 2017) - and into algorithmic fairness, where representations are encouraged to be independent of sensitive attributes (Zemel et al., 2013; Edwards & Storkey, 2015; Zhang et al., 2018; Madras et al., 2018). In parallel, deep metric learning developed geometric regularizers for discriminative structure; for example, Center Loss explicitly pulls embeddings toward their class centroids to reduce intra-class variance and sharpen separability(Wen et al., 2016).

Concurrently, an influential unsupervised line investigated whether suitable inductive biases could drive models to discover factorized structure without labels, most prominently via variational autoencoders (VAEs). $\beta$-VAE strengthens the KL term to pressure the encoder toward a factorized posterior and promote disentangling (Higgins et al., 2017). Subsequent objectives targeted dependencies more directly: $\beta$-TCVAE isolates and penalizes the total-correlation term in the ELBO (Chen et al., 2018); FactorVAE uses an adversarial density-ratio estimator to penalize total correlation (Kim & Mnih, 2018); and DIP-VAE matches moments of the aggregated posterior to a factorized prior, suppressing off-diagonal covariances (Kumar et al., 2018). In a complementary adversarial formulation, InfoGAN maximizes mutual information between a chosen subset of latent variables and the generator's output to induce interpretable factors (Chen et al., 2016).

The purely unsupervised paradigm met a foundational challenge: under broad and practically relevant conditions, disentanglement is not identifiable without appropriately tailored inductive biases. Locatello et al. formalized this limitation by showing that, in the absence of such biases, multiple equally valid latent factorizations can explain the same observations, precluding recovery of the "true" factors (Locatello et al., 2019). This impossibility result reframed the agenda from seeking a universally unsupervised solution to characterizing the precise assumptions - about models, data, or weak forms of supervision - under which disentanglement becomes well-posed.

The challenge of identifiability has been met with a range of approaches that incorporate structured assumptions or weak supervisory signals. One prominent direction shows that minimal pairwise supervision - e.g., signals about whether two observations share at least one factor or how many factors changed between them, without identifying which factors - can suffice to constrain the problem and enable disentanglement (Locatello et al., 2020a). Theoretical work has established formal conditions for identifiability, linking modern VAEs to the established theory of nonlinear ICA by showing that an observed auxiliary variable can be sufficient to provably identify the latent sources (Khemakhem et al., 2020). Framing the problem from a different information-theoretic perspective, the Disentangled Information Bottleneck seeks to use the supervisory signal to explicitly isolate and discard all information irrelevant to the supervised task, thereby learning a maximally compressed predictive representation (Pan et al., 2021). More recent efforts explore even weaker signals, using self-supervision where data augmentations serve as a proxy for interventions, providing a supervisory signal without requiring any labels (Eastwood et al., 2023).

This pursuit of identifiable representations has also spurred research into stronger structural priors, notably from the perspectives of causality and compositionality. Causal representation learning aims to align latent variables with the underlying causal mechanisms of the data, often by learning from interventional or counterfactual data, with the goal of improving robustness and out-of-distribution generalization (Schölkopf et al., 2021; Ahuja et al., 2023). In a parallel effort, object-centric models pursue compositionality by learning to parse complex scenes into a set of discrete entities, each with its own disentangled attributes. Architectures like Slot Attention and its successors thereby factorize the representation at a higher level of abstraction, moving from a flat vector of factors to a structured set of objects (Locatello et al., 2020b; Elsayed et al., 2022).

# 3 THE PRISM ARCHITECTURE

The design of the PRISM architecture is guided by a set of principles aimed at achieving a robust bipartite factorization through a carefully orchestrated set of competing objectives. The central mechanism involves imposing a tight information bottleneck on the concept-relevant subspace $\mathbf{z}_1$. By combining a standard classification objective with a structural loss that promotes intra-class compactness, we render this subspace an inefficient channel for encoding non-class variation. As our information-theoretic analysis presented in Section 4 will show, this deliberate constraint compels the model to route this necessary information into the complementary residual subspace $\mathbf{z}_0$. To ensure the $\mathbf{z}_0$ subspace remains class-agnostic, we employ an adversarial objective to purge any leaked label information - a mechanism whose efficacy we formally bound in Lemma 1. Crucially, to prevent this adversarial pressure from causing a degenerate collapse, a counter-balancing objective maximizes mutual information between $\mathbf{z}_0$ and the reconstruction. This incentivizes the residual subspace to capture the displaced intra-class variation - the core information-routing dynamic we analyze in Lemma 2. This latent-space dynamic is grounded by a dual reconstruction objective that leverages both a pixel-wise loss for structural fidelity and an adversarial loss on the image space for perceptual realism. The following subsections detail the specific loss formulations and network components that realize this principled framework.

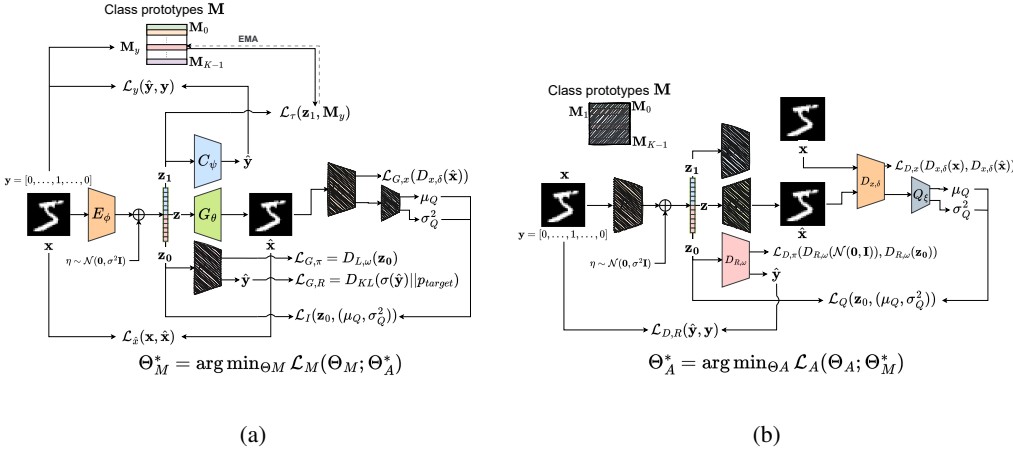

(a)                          (b)

Figure 1: The PRISM architecture and its two-player optimization framework. **(a)** The concept-extraction module ($\Theta_M$), containing the autoencoder ($E_\phi, G_\theta$) and task classifier ($C_\psi$). It is optimized via reconstruction losses ($\mathcal{L}_{\hat{x}}, \mathcal{L}_{G,x}$), losses shaping the concept subspace $\mathbf{z}_1$ ($\mathcal{L}_y, \mathcal{L}_\tau$), and losses shaping the residual subspace $\mathbf{z}_0$ ($\mathcal{L}_{G,R}, \mathcal{L}_I, \mathcal{L}_{G,\pi}$). **(b)** The adversarial module ($\Theta_A$), which provides the learning signals. It includes a multi-head latent discriminator $D_{R,\omega}$ for class prediction ($\mathcal{L}_{D,R}$) and prior matching ($\mathcal{L}_{D,\pi}$), and an image discriminator $D_{x,\delta}$ for real/fake classification ($\mathcal{L}_{D,x}$) and latent code recovery via its auxiliary head $Q_\xi$ ($\mathcal{L}_Q$).

## 3.1 ARCHITECTURAL AND OPTIMIZATION FRAMEWORK

The PRISM framework is implemented as a generative autoencoder, consisting of a stochastic encoder $E_\phi$ and a generator $G_\theta$. The encoder maps an input image $\mathbf{x}$ to a latent representation $\mathbf{z}$, which

is partitioned into a concept-relevant subspace $\mathbf{z}_1 \in \mathbb{R}^{d_1}$ and a residual subspace $\mathbf{z}_0$. The generator then reconstructs the image $\hat{\mathbf{x}}$ from the complete latent code $\mathbf{z} = (\mathbf{z}_1, \mathbf{z}_0)$.

As outlined, the training is formulated as a two-player adversarial game (Goodfellow et al., 2014). The first player, the *concept-extraction module* (Figure 1a), includes the autoencoder backbone $(E_\phi, G_\theta)$ and a task classifier $(C_\psi)$. Its parameters, denoted $\Theta_M = (\phi, \theta, \psi)$, are optimized to minimize the objective $\mathcal{L}_M$. The second player, the *adversarial module* (Figure 1b), provides the learning signals to structure the latent space. It comprises a latent-residual discriminator $(D_{R,\omega})$, an image discriminator $(D_{x,\delta})$, a mutual information estimator head $(Q_\xi)$, and a prior discriminator $(D_\pi)$. Its parameters, denoted $\Theta_A = (\omega, \delta, \xi, \pi)$, are optimized to minimize the objective $\mathcal{L}_A$.

The training objective is to find a Nash Equilibrium $(\Theta_M^*, \Theta_A^*)$ for this game. This equilibrium is the solution to the concurrent minimization problem:

$$\Theta_M^* = \arg\min_{\Theta_M} \mathcal{L}_M(\Theta_M; \Theta_A^*) \quad \text{and} \quad \Theta_A^* = \arg\min_{\Theta_A} \mathcal{L}_A(\Theta_A; \Theta_M^*) \tag{1}$$

In practice, this equilibrium is approached by alternating gradient descent steps on the two objectives, $\mathcal{L}_M$ and $\mathcal{L}_A$, with all expectations approximated over mini-batches.

## 3.2 Concept-extraction Objective ($\mathcal{L}_M$)

The objective for the concept-extraction module, $\mathcal{L}_M$, is a weighted sum of the following distinct partial loss components:

$$\mathcal{L}_M = \mathbb{E}_{(\mathbf{x},y) \sim p_{data}} \left[ \gamma_{\hat{x}} \mathcal{L}_{\hat{x}} + \gamma_x \mathcal{L}_{G,x} + \gamma_y \mathcal{L}_y + \gamma_\tau \mathcal{L}_\tau + \gamma_L \mathcal{L}_{G,R} + \gamma_I \mathcal{L}_I + \gamma_\pi \mathcal{L}_{G,\pi} \right] \tag{2}$$

where the $\gamma_* \geq 0$ are nonnegative weighting coefficients.

**Reconstruction Fidelity.** Two terms enforce high-quality image synthesis. The reconstruction loss, $\mathcal{L}_{\hat{x}}$, estimates similarity between the input $\mathbf{x}$ and the Generator's output $\hat{\mathbf{x}}$. The specific form of this loss is adapted to the data complexity; we use a pixel-wise Mean Squared Error (MSE) for simpler datasets, and a VGG-based perceptual loss for complex, natural images to better capture textural and semantic details (Johnson et al., 2016). This is complemented by an adversarial loss, $\mathcal{L}_{G,x}$, which drives the generator to produce images that are indistinguishable from real data to the image discriminator $D_{x,\delta}$. Our implementation uses a non-saturating GAN objective (Goodfellow et al., 2014).

**Structuring the Concept-Relevant Subspace ($\mathbf{z}_1$).** The $\mathbf{z}_1$ is shaped by two loss components. A standard classification loss, $\mathcal{L}_y$, ensures that $\mathbf{z}_1$ encodes class-discriminative features, and is defined as the cross-entropy between the classifier's output $C_\psi(\mathbf{z}_1)$ and the true label $\mathbf{y}$: $\mathcal{L}_y = \mathcal{L}_{CE}(C_\psi(\mathbf{z}_1), \mathbf{y})$. A structural loss, $\mathcal{L}_\tau = \|\mathbf{z}_1 - \mathbf{M}_y\|_2^2$, promotes intra-class compactness by pulling embeddings towards learned class prototypes $\mathbf{M}_y$. To stabilize these prototype targets, we update $\mathbf{M}_y$ using an Exponential Moving Average (EMA) of batch-wise class means. This approach, common for preventing target network instability, ensures the prototypes serve as slowly-evolving targets for the encoder (Tarvainen & Valpola, 2017; He et al., 2020; Grill et al., 2020). Specifically, we update the prototype for any class $y$ that appears in the current minibatch via

$$\mathbf{M}_y \leftarrow m\,\mathbf{M}_y + (1 - m)\,\widehat{\boldsymbol{\mu}}_y, \tag{3}$$

where $m \in [0, 1)$ denotes the momentum term, $\widehat{\boldsymbol{\mu}}_y$ is the batch mean of $\mathbf{z}_1$ codes for samples corresponding to a label $y$; prototypes for classes not represented in a batch remain unchanged.

**Shaping the Residual Subspace ($\mathbf{z}_0$).** The residual subspace $\mathbf{z}_0$ is engineered to be both class-agnostic and informative for reconstruction. First, the encoder is trained to fool the latent adversarial classifier via the loss $\mathcal{L}_{G,R}$, which minimizes information on a class label $y$ embedded in $\mathbf{z}_0$. This is formulated as the KL divergence between the adversarial classifier's predictions, normalized by the softmax function $\sigma(\cdot)$, and a fixed target distribution $p_{\text{target}}$: $\mathcal{L}_{G,R} = D_{KL}(\sigma(D_{R,\omega}(\mathbf{z}_0)) \,\|\, p_{\text{target}})$. Second, to ensure $\mathbf{z}_0$ retains information necessary for reconstruction, we maximize a variational lower bound on the mutual information $I(\mathbf{z}_0; \hat{\mathbf{x}})$ using the loss term $\mathcal{L}_I$. We use a standard variational MI objective with an auxiliary predictor and a reference prior (see section A.2.2). The bound tightens as the aggregated posterior approaches the prior; we encourage this via the prior-matching loss $\mathcal{L}_{G,\pi}$, implemented as a generator-side GAN objective for the prior discriminator $D_\pi$ (Makhzani et al., 2015) (in practice a dedicated head on the multi-task latent-residual discriminator $D_{R,\omega}$ performs this role).

## 3.3 ADVERSARIAL NETWORK OBJECTIVE ($\mathcal{L}_A$)

Concurrently, the four auxiliary networks are trained to minimize the adversarial loss $\mathcal{L}_A$:

$$\mathcal{L}_A = \mathbb{E}_{(\mathbf{x},y) \sim p_{data}} \left[ \gamma_L \mathcal{L}_{D,R} + \gamma_x \mathcal{L}_{D,x} + \gamma_I \mathcal{L}_Q + \gamma_\pi \mathcal{L}_{D,\pi} \right] + \mathcal{L}_{R1} \tag{4}$$

where, as above, the $\gamma_* \geq 0$ are nonnegative weighting coefficients. The latent-residual classifier $D_{R,\omega}$ is trained via a standard cross-entropy loss, $\mathcal{L}_{D,R} = \mathcal{L}_{CE}(D_{R,\omega}(\mathbf{z}_0), \mathbf{y})$, to accurately predict $y$ from $\mathbf{z}_0$. The image discriminator $D_{x,\delta}$ is trained via $\mathcal{L}_{D,x}$ to distinguish real images from generated ones, using non-saturating GAN objective. Its training is stabilized with an R1 gradient penalty on real data: $\mathcal{L}_{R1} = \frac{\gamma_{R1}}{2} \mathbb{E}_{\mathbf{x}}[\|\nabla_{\mathbf{x}} D_{x,\delta}(\mathbf{x})\|_2^2]$ (Mescheder et al., 2018). The prior discriminator $D_\pi$ is trained via $\mathcal{L}_{D,\pi}$ to distinguish between samples from the encoder's aggregated posterior $q_\phi(\mathbf{z}_0)$ and the fixed prior $r(\mathbf{z}_0)$, using a standard GAN objective. Finally, the Q-network head $Q_\xi$ is trained via a negative log-likelihood loss, $\mathcal{L}_Q$, to recognize the specific code $\mathbf{z}_0$ used to generate an image $\hat{\mathbf{x}}$. This trains $Q_\xi$ to approximate the posterior $p(\mathbf{z}_0 \mid \hat{\mathbf{x}})$, enabling the information-maximization objective $\mathcal{L}_I$.

# 4 A PRINCIPLED FRAMEWORK FOR INFORMATION PARTITIONING

This section formalizes the information-theoretic principles that motivate the PRISM architecture. We demonstrate how the model's objectives are designed to encourage a robust partition of its latent space. Our analysis explores the properties of this partition at a hypothetical equilibrium point.

## 4.1 THEORETICAL PRELIMINARIES AND SCOPE

Our theoretical analysis is conditioned on the strong assumption that the complex adversarial training converges to a stable equilibrium where the population-level errors are bounded. While convergence is not guaranteed in practice for multi-agent optimization, this analysis provides insights into the model's intended dynamics at such an equilibrium. To proceed with the derivation of key properties of the proposed solution, we formulate a set of realistic assumptions, detailed in section A.2. In particular, we assume sufficient information capacity of all networks involved in data processing and boundedness of population errors for the latent adversarial game, the reconstruction loss, and the structural loss.

## 4.2 ADVERSARIAL SUPPRESSION OF CLASS INFORMATION

Our approach for structuring the residual subspace $\mathbf{z}_0$ can be viewed through the lens of the Information Bottleneck (IB) principle (Tishby et al., 1999). This framework seeks a representation that is minimally informative about a nuisance variable (the class label $y$) while retaining maximal information about a target variable (the reconstruction $\hat{\mathbf{x}}$). This can be expressed via the Lagrangian:

$$\mathcal{J}(E_\phi) = I(y; \mathbf{z}_0) - \beta \, I(\mathbf{z}_0; \hat{\mathbf{x}}). \tag{5}$$

As both mutual information terms are intractable to optimize directly, PRISM employs tractable surrogates for each, similar in spirit to prior work in Adversarial Information Bottleneck (Zhai & Zhang, 2022).

**Minimizing $I(y; \mathbf{z}_0)$ via adversarial training.** The encoder is trained to minimize the adversarial loss $\mathcal{L}_{G,R}$, which encourages it to produce $\mathbf{z}_0$ embeddings that the latent-residual discriminator $D_{R,\omega}$ cannot distinguish. By the Data Processing Inequality for $y \to \mathbf{z}_0 \to \hat{y}_L$, we have $I(y; \hat{y}_L) \leq I(y; \mathbf{z}_0)$. We therefore minimize the *surrogate leakage* $I(y; \hat{y}_L)$ by pushing the discriminator's prediction $\sigma(D_{R,\omega}(\mathbf{z}_0))$ toward a target distribution $p_{\text{target}}$ (uniform or $p(y)$; see section A.3.2), i.e., making $\hat{y}_L$ uninformative about $y$. In the adversarial game, any $y$-signal in $\mathbf{z}_0$ that the discriminator can exploit penalizes the encoder; forcing $\sigma(D_{R,\omega}(\mathbf{z}_0))$ to be uninformative, thus encouraging the encoder to remove label information from $\mathbf{z}_0$ - reducing label leakage into $\mathbf{z}_0$.

The intended mechanism for purging class information from the residual subspace is formalized in the following lemma, which bounds the information leakage at a theoretical equilibrium.

**Lemma 1** (Upper Bound on Information Leakage). *Under the stated assumptions, the mutual information between the class label $y$ and the residual code $\mathbf{z}_0$ is bounded by:*

$$I(y; \mathbf{z}_0) \ \leq \ \frac{4}{p_{\min}}\big(\varepsilon_D + \eta\big), \tag{6}$$

*where $p_{\min}$ is the minimum class probability, and $\varepsilon_D$ and $\eta$ bound, respectively, the expected KL divergence from the true posterior $p(y \mid \mathbf{z}_0)$ to the discriminator $q_\omega(y \mid \mathbf{z}_0)$ and from the discriminator to the class prior $p(y)$, with $q_\omega$ learned by $D_{R,\omega}$. (Proof in section A.3).*

This result provides a quantitative link between the adversarial game's equilibrium errors $(\varepsilon_D, \eta)$ and the degree of class-independence that would be achieved at this equilibrium.

**Maximizing $I(\mathbf{z}_0; \hat{\mathbf{x}})$ with an InfoGAN-style objective and AAE-style marginal alignment.** To keep $\mathbf{z}_0$ useful for reconstruction, we introduce an auxiliary predictor $Q_\xi(\mathbf{z}_0 \mid \hat{\mathbf{x}})$ and maximize a tractable InfoGAN-style lower bound on $I(\mathbf{z}_0; \hat{\mathbf{x}})$ (Chen et al., 2016). The decomposition and training implications are stated in Proposition 1: the gap to the true mutual information is the conditional divergence between the true posterior and $Q_\xi$, which we reduce by training $Q_\xi$ to recover the actual residual code from reconstructions. In tandem, an Adversarial Autoencoder–style prior-matching loss aligns the aggregated residual distribution $q_\phi(\mathbf{z}_0)$ with a fixed prior $r(\mathbf{z}_0)$ (Makhzani et al., 2015). For a fixed encoder and $Q_\xi$, this leaves the gap unchanged and raises the bound by shrinking the marginal mismatch.

### 4.3 Routing of Intra-Class Variation via Structural Constraints

While the Information Bottleneck framework describes the direct objectives applied to $\mathbf{z}_0$, a key design principle of our model is that constraints on one part of the latent space should incentivize the utilization of another. The objective of high-fidelity reconstruction creates a "pull" for information, requiring the full latent code $\mathbf{z}$ to capture sufficient information about the input $\mathbf{x}$. Concurrently, the structural and classification losses on $\mathbf{z}_1$ create a "push" by severely restricting its information capacity, limiting it to primarily class-relevant features. This renders $\mathbf{z}_1$ an inefficient channel for the rich, intra-class variation required for a faithful reconstruction. Consequently, the encoder is incentivized to route, or displace, this residual information into the less constrained subspace, $\mathbf{z}_0$.

This emergent dynamic, where information is necessarily channeled into the residual subspace, is formalized in the following lemma.

**Lemma 2** (Lower Bound on Conditional Residual Information). *Under the stated assumptions, the information about the input $\mathbf{x}$ that is uniquely available in the residual subspace $\mathbf{z}_0$ is lower-bounded by:*

$$\boxed{I(\mathbf{x}; \mathbf{z}_0 \mid \mathbf{z}_1) \ \geq \ R_X(\delta_{rec}) - H(y) - \frac{d_1}{2}\log\Big(\frac{\delta_\tau}{d_1\sigma^2}\Big)} \tag{7}$$

(Proof in section A.4).

Lemma 2 formalizes the consequences of this mechanism. It establishes a clear theoretical relationship between reconstruction fidelity and the information content of the residual subspace. The bound identifies three primary architectural levers likely to control the amount of information preserved in $\mathbf{z}_0$. It predicts that the information content of $\mathbf{z}_0$ can be increased by: (i) improving reconstruction fidelity (decreasing $\delta_{rec}$), thereby increasing the total information demand $R_X(\delta_{rec})$ (where $R_X(\cdot)$ is the Shannon rate-distortion function; see Appendix A.4.1 for a formal definition); (ii) reducing the dimensionality of the task-relevant subspace, $d_1$; or (iii) tightening the intra-class compactness of $\mathbf{z}_1$ (decreasing $\delta_\tau$). While the rate-distortion function $R_X(\delta_{rec})$ is intractable, the lemma's primary utility lies in its prescriptive power. It provides clear, falsifiable hypotheses about the causal effects of these architectural choices, which we verify empirically in our experiments (see section 5.3).

## 5 Experimental evaluation

### 5.1 Datasets

We use two datasets for the experimental evaluation. First, we conduct an ablation study and validate the hypothesis emerging from the formulation of Lemma 2 using the Morpho-MNIST (Castro et al.,

2019) dataset - an extension of MNIST (Lecun et al., 1998) that introduces a set of quantifiable perturbations and morphometric labels. We use the variant that applies thinning and thickening to the digits, thereby increasing intra-class variation. We adopt the train/test split provided by the authors and reserve 10,000 images from the training set for validation. For qualitative experiments, we use the CelebA dataset, which contains 202,599 face images of 10,177 identities and 40 binary attributes; we follow the official train/val/test split, center-crop to $178 \times 178$, and resize to $128 \times 128$.

## 5.2 ABLATION STUDY

To assess the influence of each PRISM component, we perform an ablation study using a two-stage protocol. In stage one, each configuration receives the same *hyperparameter search budget* and is optimized for the probe-gap metric (measured in percentage points), defined as the difference between the accuracies of the two linear probes (regularized linear classifiers trained post hoc on standardized features): one trained to predict the label $y$ from the concept-relevant subspace $\mathbf{z}_1$ and the other from the residual subspace $\mathbf{z}_0$. In stage two, the best setting for each configuration is retrained with five independent seeds, with all hyperparameters fixed. All metrics are averaged across the five seeds and reported as mean $\pm$ s.d. To further characterize the subspaces, we cluster embeddings from each subspace independently using $k$-means with $k$ equal to the number of classes ($k = 10$ for Morpho-MNIST), and compute the adjusted Rand index (ARI) (Hubert & Arabie, 1985) and silhouette score (Rousseeuw, 1987) for each clustering.

Table 1: Ablation study results. Higher is better for probe-gap, $\mathbf{z}_1$ ARI, and $\mathbf{z}_1$ silhouette score; lower is better for $\mathbf{z}_0$ ARI.

| configuration | probe-gap (pp) ↑ | $\mathbf{z}_1$ ARI ↑ | $\mathbf{z}_1$ sil. score ↑ | $\mathbf{z}_0$ ARI ↓ |
|---|---|---|---|---|
| Baseline ($\mathcal{L}_{\hat{x}}, \mathcal{L}_{G,x}$) | $0.56 \pm 6.83$ | $0.23 \pm 0.05$ | $0.15 \pm 0.00$ | $0.18 \pm 0.03$ |
| + $\mathcal{L}_y$ | $48.55 \pm 2.74$ | $0.91 \pm 0.01$ | $0.35 \pm 0.01$ | $0.07 \pm 0.02$ |
| + $\mathcal{L}_\tau$ | $20.53 \pm 7.85$ | $0.14 \pm 0.03$ | $0.13 \pm 0.01$ | $0.05 \pm 0.03$ |
| + $\mathcal{L}_{G,R}$ | $59.37 \pm 2.47$ | $0.42 \pm 0.02$ | $0.18 \pm 0.01$ | $0.05 \pm 0.02$ |
| + ($\mathcal{L}_I, \mathcal{L}_{G,\pi}$) | $35.71 \pm 3.05$ | $0.21 \pm 0.02$ | $0.16 \pm 0.01$ | $0.06 \pm 0.02$ |
| + $\mathcal{L}_y + \mathcal{L}_\tau$ | $41.19 \pm 1.49$ | $0.81 \pm 0.05$ | $0.33 \pm 0.02$ | $0.08 \pm 0.01$ |
| + $\mathcal{L}_{G,R} + (\mathcal{L}_I, \mathcal{L}_{G,\pi})$ | $45.02 \pm 6.81$ | $0.29 \pm 0.04$ | $0.17 \pm 0.00$ | $0.09 \pm 0.03$ |
| + $\mathcal{L}_y + \mathcal{L}_{G,R}$ | $74.31 \pm 4.77$ | $0.93 \pm 0.05$ | $0.43 \pm 0.03$ | $0.03 \pm 0.02$ |
| + $\mathcal{L}_y + \mathcal{L}_{G,R} + \mathcal{L}_\tau$ | $72.40 \pm 2.09$ | $\mathbf{0.96} \pm 0.03$ | $\mathbf{0.61} \pm 0.03$ | $0.03 \pm 0.01$ |
| + $\mathcal{L}_y + \mathcal{L}_{G,R} + (\mathcal{L}_I, \mathcal{L}_{G,\pi})$ | $70.02 \pm 2.33$ | $0.74 \pm 0.12$ | $0.31 \pm 0.03$ | $0.04 \pm 0.01$ |
| Full Model | $\mathbf{75.50} \pm 1.56$ | $0.90 \pm 0.06$ | $0.40 \pm 0.02$ | $\mathbf{0.01} \pm 0.00$ |

As shown in Table 1, the full model achieves the highest probe-gap (75.50 pp), while the variant without $\mathcal{L}_\tau$ and ($\mathcal{L}_I, \mathcal{L}_{G,\pi}$) is a close contender (74.31 pp). Neither the classifier alone nor the latent-residual adversary alone is sufficient to enforce a clear, bipartite factorization of the latent space. Label alignment - measured by $\mathbf{z}_1$ ARI - and cluster intra-class compactness and inter-class separation - measured by the $\mathbf{z}_1$ silhouette score - are the highest when the prototype regularizer $\mathcal{L}_\tau$ is included alongside $\mathcal{L}_y$ and $\mathcal{L}_{G,R}$, confirming its intended role in structuring the concept-relevant subspace $\mathbf{z}_1$. Although the ($\mathcal{L}_I, \mathcal{L}_{G,\pi}$) term, regularizing the residual subspace $\mathbf{z}_0$, has a detrimental influence on these factors, its addition leads to the lowest $\mathbf{z}_0$ ARI result, clearly indicating a weaker label signal in the residual subspace $\mathbf{z}_0$ and contributing to the overall success of the full model. A quantitative comparison against the Fader Network baseline is provided in section 5.4. The detailed plots showing the latent structures and the qualitative results for Morpho-MNIST are provided in section C.

## 5.3 EVALUATION OF LEMMA 2 PREDICTIONS

We conduct a series of controlled experiments to empirically validate the key predictions of Lemma 2. To quantify the information routed into the residual subspace $\mathbf{z}_0$, we require a tractable proxy for the conditional mutual information term $I(\mathbf{x}; \mathbf{z}_0 \mid \mathbf{z}_1)$. We adopt the informativeness score from the DCI framework (Eastwood & Williams, 2018) for this purpose, calculated exclusively on the residual subspace $\mathbf{z}_0$. This metric evaluates how accurately a simple model - in this case, a random forest - can predict the ground-truth generative factors from the latent representations. Since

the model is trained to route concept-relevant information into $\mathbf{z}_1$, the informativeness of $\mathbf{z}_0$ with respect to the remaining (non-class) factors serves as a strong surrogate for the amount of intra-class variation captured in the residual space. This directly corresponds to the quantity $I(\mathbf{x}; \mathbf{z}_0 \mid \mathbf{z}_1)$ that Lemma 2 aims to characterize. To test the lemma's predictions, we systematically manipulate the three architectural levers it identifies. For each experiment, all model variants were trained five times with different random seeds to ensure robustness. The results are reported as the mean DCI informativeness, with the shaded regions representing the 95% confidence interval across runs.

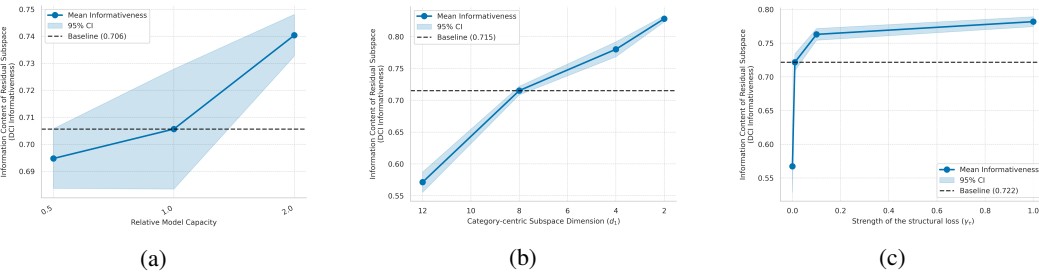

|  (a) | (b) | (c) |

Figure 2: Validation of the information-routing dynamics predicted by Lemma 2. The DCI informativeness of the residual subspace $\mathbf{z}_0$ is shown to be a function of: (a) model capacity (proxy for reconstruction fidelity), (b) concept subspace dimensionality ($d_1$), and (c) the strength of the structural loss ($\gamma_\tau$).

Our validation begins by examining the influence of reconstruction fidelity, which the lemma predicts will control the total information demand $R_X(\delta_{rec})$. To test this, we modulate the model's ability to reconstruct inputs by varying the information capacity of the autoencoder architecture, creating variants with 50%, 100% (baseline), and 200% of the original capacity. The results, shown in Figure 2a, are consistent with the prediction that lower fidelity reduces the information routed into $\mathbf{z}_0$, revealing a positive trend where higher model capacity corresponds to greater informativeness in the residual subspace. Beyond the total information preserved, the lemma also makes specific predictions about how this information is partitioned. We therefore isolated the effect of the concept subspace dimensionality, $d_1$, by training models with $d_1$ set to 12, 8, 4, and 2. Crucially, all variants were configured to achieve a nearly-identical reconstruction error $\delta_{rec}$, thereby holding the total information demand $R_X(\delta_{rec})$ approximately constant. Figure 2b shows a strong inverse correlation between the dimensionality of the concept subspace and the informativeness of the residual subspace. As the information-carrying capacity of $\mathbf{z}_1$ is reduced, the encoder is compelled to route a greater amount of variation through $\mathbf{z}_0$. The final lever for controlling this information partition is the structural constraint imposed on $\mathbf{z}_1$. We tested the prediction that tightening this constraint via the structural loss weight, $\gamma_\tau$, displaces information into $\mathbf{z}_0$. As in the previous experiment, reconstruction error was held nearly constant across all runs. As shown in Figure 2c, increasing the strength of this constraint consistently increases the information content of the residual subspace. The effect exhibits the logarithmic characteristic predicted by the $-\frac{d_1}{2}\log(\cdot)$ term in Lemma 2, with diminishing returns as the structural loss becomes stronger. Taken together, the consistent outcomes across these three distinct manipulations provide robust empirical support for the information-routing dynamics formalized in our theoretical framework.

## 5.4 QUANTITATIVE RESULTS

To further contextualize the performance of PRISM, we conduct a direct comparison with Fader Networks (Lample et al., 2017), a well-established baseline for supervised disentanglement. To ensure a fair comparison, we configured the Fader Network to use the identical encoder-decoder backbone as our PRISM model on the Morpho-MNIST dataset (see section B.1). The key difference lies in the disentanglement mechanism: Fader Networks apply an adversarial classifier to the entire latent space to make it invariant to the supervised attribute, whereas PRISM explicitly partitions the latent space and applies adversarial pressure only to the residual subspace $\mathbf{z}_0$. We evaluate both models on reconstruction fidelity (SSIM; (Wang et al., 2004)), disentanglement quality (DCI; (Eastwood & Williams, 2018) and SAP Score; (Kumar et al., 2018)), and the degree of information

leakage (Linear Probe Accuracy), with results averaged over five independent runs and summarized in Table 2.

Table 2: Quantitative comparison of PRISM and Fader Networks on Morpho-MNIST. For PRISM, DCI, SAP, and the linear probe are evaluated exclusively on the residual subspace $z_0$. For Fader Networks, they are evaluated on the full latent space $z$. Results are mean $\pm$ s.d. over five runs.

| Metric | PRISM (Ours) | Fader Network |
|---|---|---|
| DCI Informativeness $\uparrow$ | $0.689 \pm 0.020$ | $\mathbf{0.713 \pm 0.020}$ |
| DCI Disentanglement $\uparrow$ | $0.206 \pm 0.050$ | $\mathbf{0.257 \pm 0.023}$ |
| DCI Completeness $\uparrow$ | $0.274 \pm 0.059$ | $\mathbf{0.314 \pm 0.025}$ |
| SAP Score $\uparrow$ | $\mathbf{0.123 \pm 0.055}$ | $0.053 \pm 0.025$ |
| SSIM $\uparrow$ | $\mathbf{0.806 \pm 0.007}$ | $0.790 \pm 0.005$ |
| Label Leakage (Probe Acc.) $\downarrow$ | $\mathbf{23.96\% \pm 2.33}$ | $25.76\% \pm 2.07$ |

The results in Table 2 validate the efficacy of PRISM in achieving the objective of bipartite factorization. First, regarding information preservation, both models achieve high DCI Informativeness scores (0.689 for PRISM vs. 0.713 for Fader Networks). This confirms that PRISM's residual subspace $z_0$ successfully retains the necessary stylistic factors (e.g., thickness, slant) required for reconstruction, empirically supporting the information-routing mechanism described in Lemma 2. The comparable scores indicate that the explicit partitioning and bottleneck on $z_1$ do not result in a significant loss of generative content in $z_0$ relative to the unpartitioned baseline.

The metrics further reveal a divergence in the structural organization of this retained information. Fader Networks achieve higher DCI Disentanglement scores. As DCI is computed using non-linear predictors (Random Forests), it is robust to distributed representations where factors of variation are encoded by subsets of latent dimensions. In contrast, PRISM achieves a higher SAP score (0.123 vs 0.053). This result indicates that PRISM's residual dimensions are more axis-aligned with the generative factors, as SAP explicitly rewards representations where single latent dimensions dominate the prediction of specific factors. This alignment is a direct consequence of the prior-matching loss $\mathcal{L}_{D,\pi}$, which minimizes the KL divergence between the aggregated posterior $q_\phi(z_0)$ and a factorized isotropic Gaussian prior, thereby encouraging statistical independence among the residual coordinates.

Crucially, PRISM achieves this structured factorization while optimizing the trade-off between attribute invariance and reconstruction quality. PRISM demonstrates lower label leakage (23.96%) compared to Fader Networks (25.76%), indicating a more effective removal of the supervised attribute from the residual space. Simultaneously, PRISM attains an improved SSIM score (0.806 vs 0.790). This implies that the explicit subspace partitioning allows the model to route attribute information out of $z_0$ without compromising the high-frequency details required for faithful synthesis, whereas the baseline's adversarial penalty on the full latent space induces a greater compromise on reconstruction fidelity.

## 5.5 QUALITATIVE RESULTS

To complement the quantitative validation of our theoretical framework, we now turn to a qualitative interrogation of the learned representations on the high-dimensional CelebA dataset. These experiments are designed to visually assess the bipartite factorization and demonstrate its utility for fine-grained semantic control by operating independently on the two latent subspaces.

First, we verify that the model can perform targeted attribute swapping. As shown in Figure 3a and 3b exchanging the residual codes $z_0$ between subjects cleanly transfers the targeted attributes of gender and smile, while preserving the identity and style of the original image. This provides strong evidence that the supervised information, isolated within the concept subspace $z_1$, is effectively separated from the stylistic attributes contained in $z_0$. Next, we probe the semantic roles of each subspace more directly. By replacing the residual code with its dataset-average, we can generate a "canonical" version of a concept, effectively normalizing stylistic variation (Figure 3c and 3d middle rows). Conversely, replacing the concept code with its average evens out the target attribute while preserving the subject's unique appearance (bottom rows). Finally, traversing the principal

components of $\mathbf{z}_0$ reveals a smooth and meaningful representation of class-agnostic features like background color and head pose (Figure 3e), confirming it has not collapsed to a degenerate solution.

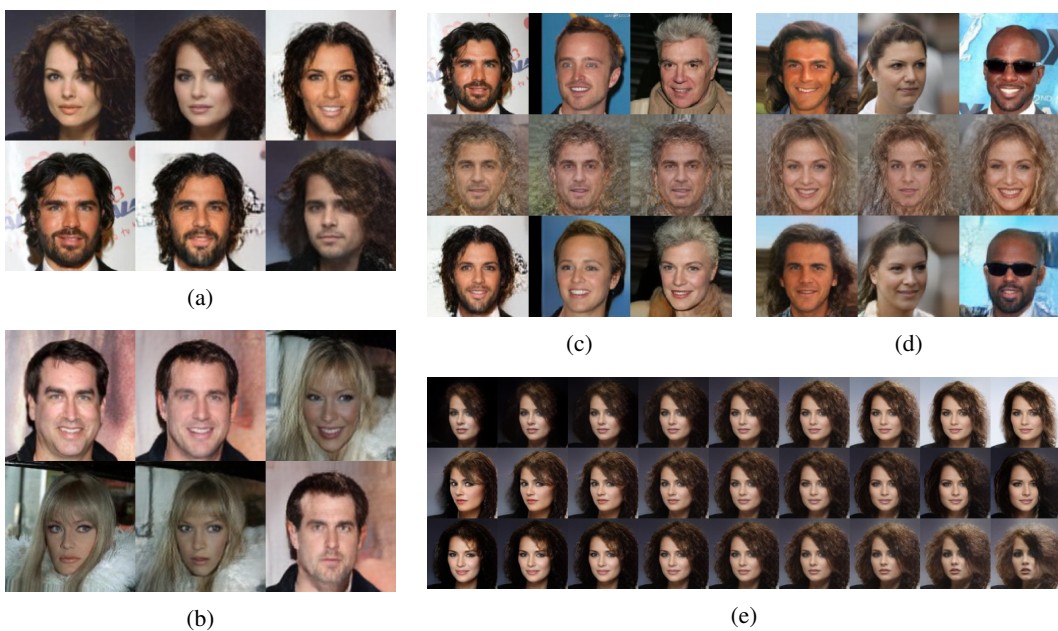

(a)

(c)

(d)

(b)

(e)

Figure 3: **Semantic attribute control on CelebA. (a, b)** Attribute swapping is achieved by combining the concept code $\mathbf{z}_1$ from a source image (top row) with the residual code $\mathbf{z}_0$ from a reference image (bottom row). The result (right column) preserves the supervised attribute—(a) gender and (b) smile—from the source while adopting the style and identity of the reference. **(c, d)** Mean code replacement for models disentangling (c) gender and (d) smile. Rows show: the original sample, its reconstruction with mean $\mathbf{z}_0$, and its reconstruction with mean $\mathbf{z}_1$. **(e)** Latent traversal along the top three principal components of the residual subspace $\mathbf{z}_0$.

## 6 CONCLUSIONS

This work introduces a principled, adversarial approach to bipartite latent space factorization. The presented learning mechanism enables effective separation of factors inherent to known (labeled) concepts from concept-irrelevant information into two separate latent subspaces: concept-related and residual. We provided a theoretical framework for this process, establishing an upper bound on information leakage into the residual subspace and identifying key architectural levers that drive the routing of intra-class variation into this subspace. These factors were confirmed in our empirical evaluation.

We have also demonstrated the effectiveness of the proposed bipartite latent space factorization by performing style swapping experiments on both toy and real-world datasets. Beyond these immediate capabilities, the PRISM architecture offers a distinct advantage over standard conditional generative models: by learning a continuous concept subspace $\mathbf{z}_1$ rather than conditioning on fixed one-hot vectors, the model produces a purpose-built feature embedding. This opens several promising directions for future work. For example, the purified, continuous representation of concepts can become an invaluable basis for input contents explainability, exploiting information distilled from factors of variation. Alternatively, by focusing the analysis on the residual subspace, one can try to identify and assess the importance of different factors of variation entangled with real-world concept categories.

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

# A  THEORETICAL JUSTIFICATION FOR THE PRISM ARCHITECTURE

## A.1  IDEALIZED EQUILIBRIUM: A THOUGHT EXPERIMENT

To clarify the intended role of each architectural component, we consider an idealized thought experiment. We analyze the properties of the latent space at a hypothetical equilibrium point where all non-adversarial loss terms are zero (i.e., $\mathcal{L}_{\hat{x}} = \mathcal{L}_y = \mathcal{L}_\tau = 0$) and all adversarial games have reached a perfect Nash Equilibria. In this theoretical state, the structural loss condition ($\mathcal{L}_\tau = 0$) implies that the concept-relevant code $\mathbf{z}_1$ collapses to a class-specific prototype, becoming a deterministic function of the label $y$. Concurrently, a perfect latent adversarial equilibrium implies that $p(y \mid \mathbf{z}_0) = p(y)$, making the residual code $\mathbf{z}_0$ statistically independent of the class label. Finally, the perfect reconstruction condition ($\mathcal{L}_{\hat{x}} = 0$) necessitates that $\mathbf{z}_0$ must encode all remaining non-class information required to perfectly reconstruct the specific instance $\mathbf{x}$. This idealized state illustrates the model's core inductive bias: to partition the latent space into a subspace encoding only the class label and a residual subspace that is both class-independent and carries all other factors of variation.

## A.2  FINITE-ERROR ANALYSIS: ASSUMPTIONS AND CONDITIONS

Our formal proofs are built upon a finite-error framework that assumes the model has converged to an effective, but not perfect, equilibrium. This requires a set of standard technical conditions and a set of bounded-error assumptions that characterize the performance of a well-trained model.

### A.2.1  TECHNICAL CONDITIONS

- **(T1) Stochastic Encoder:** The encoder is of the form $\mathbf{z} = f_\phi(\mathbf{x}) + \boldsymbol{\eta}$, where the noise $\boldsymbol{\eta} \sim \mathcal{N}(\mathbf{0}, \sigma^2 \mathbf{I})$ is additive, isotropic Gaussian noise independent of the input $\mathbf{x}$.
- **(T2) Well-Defined Markov Chain:** The generator defines a conditional density $p_\theta(\hat{\mathbf{x}} \mid \mathbf{z})$, establishing the Markov chain $\mathbf{x} \to \mathbf{z} \to \hat{\mathbf{x}}$.
- **(T3) Discriminator Regularity:** The discriminator's predictive distribution $q_\omega(y \mid \mathbf{z}_0)$ has full support for all classes.
- **(T4) Sufficient Network Capacity:** The function classes for all networks are sufficiently expressive to achieve the bounded errors defined below.

**Note on (T1).** Assumption (T1) models the encoder with fixed-variance additive noise, which is a simplification over architectures that learn an input-dependent variance. This choice is made for analytical tractability, as it ensures the conditional entropy term $h(\mathbf{z}_1 \mid \mathbf{x})$ is a data-independent constant (Cover & Thomas, 2006), leading to a clean, interpretable bound in Lemma 2.

### A.2.2  FINITE-ERROR ASSUMPTIONS

We replace the zero-error idealization with the following bounded-error assumptions.

- **(B1) $\varepsilon_D$-Suboptimal Adversary:** The latent-residual discriminator $D_{R,\omega}$ converges to parameters $\omega$ such that its expected KL divergence from the true posterior is bounded:
$$\mathbb{E}_{\mathbf{z}_0 \sim q_\phi(\mathbf{z}_0)}\big[D_{KL}\big(p(y \mid \mathbf{z}_0) \,\big\|\, q_\omega(y \mid \mathbf{z}_0)\big)\big] \leq \varepsilon_D. \tag{8}$$
- **(B2) $\eta$-Approximate Encoder Objective Minimization:** The encoder $E_\phi$ minimizes its adversarial objective $\mathcal{L}_{G,R}$ to within a bounded error $\eta$:
$$\mathbb{E}_{\mathbf{z}_0 \sim q_\phi(\mathbf{z}_0)}\big[D_{KL}\big(q_\omega(y \mid \mathbf{z}_0) \,\big\|\, p(y)\big)\big] \leq \eta. \tag{9}$$
- **(B3) $\delta_{rec}$-Bounded Reconstruction Error:** The autoencoder has a population mean-squared reconstruction error bounded by $\delta_{rec}$.
- **(B4) $\delta_\tau$-Bounded Structural Loss:** The population structural loss is bounded by $\delta_\tau$.
- **(B5) Well-Posed Quantities:** All relevant moments and entropies exist and are finite.
- **(B6) Full Support of Class Prior:** The true marginal class distribution $p(y)$ has a minimum class probability $p_{\min} > 0$.

- **(B7) Finite Rate-Distortion:** For the error $\delta_{rec}$ from (B3), the Shannon rate-distortion function $R_X(\delta_{rec})$ is finite.

- **(B8) $\varepsilon_\pi$-Bounded Prior Mismatch** *(so that $D_{KL}$ is well-defined, which implies $q_\phi \ll r$):* The aggregated posterior of the residual subspace, $q_\phi(\mathbf{z}_0)$, is assumed to be close to the fixed prior $r(\mathbf{z}_0)$, with a bounded Kullback–Leibler (KL) divergence:

$$D_{KL}\big(q_\phi(\mathbf{z}_0) \,\|\, r(\mathbf{z}_0)\big) \leq \varepsilon_\pi. \tag{10}$$

**Proposition 1** (Barber–Agakov with marginal mismatch). *For any variational posterior $Q_\xi(\mathbf{z}_0 \mid \hat{\mathbf{x}})$ and prior $r(\mathbf{z}_0)$, the mutual information $I(\mathbf{z}_0; \hat{\mathbf{x}})$ is lower-bounded by a tractable objective $\mathcal{L}_I$ less a penalty for the mismatch between the aggregated posterior and the prior (Barber & Agakov, 2004):*

$$I(\mathbf{z}_0; \hat{\mathbf{x}}) \;\geq\; \underbrace{\mathbb{E}_{q_\phi(\mathbf{z}_0, \hat{\mathbf{x}})}[\log Q_\xi(\mathbf{z}_0 \mid \hat{\mathbf{x}}) - \log r(\mathbf{z}_0)]}_{\mathcal{L}_I} \;-\; D_{KL}\big(q_\phi(\mathbf{z}_0) \,\|\, r(\mathbf{z}_0)\big). \tag{11}$$

*Proof.* The derivation begins with the definition of mutual information, $I(\mathbf{z}_0; \hat{\mathbf{x}}) = \mathbb{E}_{q_\phi}[\log q_\phi(\mathbf{z}_0 \mid \hat{\mathbf{x}}) - \log q_\phi(\mathbf{z}_0)]$ (Cover & Thomas, 2006). We introduce the variational posterior $Q_\xi(\mathbf{z}_0 \mid \hat{\mathbf{x}})$ and the prior $r(\mathbf{z}_0)$ by adding and subtracting their log-densities. Regrouping the terms yields an exact identity:

$$I(\mathbf{z}_0; \hat{\mathbf{x}}) = \mathcal{L}_I + \mathbb{E}_{q_\phi(\hat{\mathbf{x}})}\big[D_{KL}(q_\phi(\mathbf{z}_0 \mid \hat{\mathbf{x}}) \,\|\, Q_\xi(\mathbf{z}_0 \mid \hat{\mathbf{x}}))\big] - D_{KL}(q_\phi(\mathbf{z}_0) \,\|\, r(\mathbf{z}_0)). \tag{12}$$

The second term, representing the average approximation error of the variational posterior, is non-negative. Dropping this term yields the lower bound. Equality holds iff $Q_\xi(\mathbf{z}_0 \mid \hat{\mathbf{x}}) = q_\phi(\mathbf{z}_0 \mid \hat{\mathbf{x}})$ almost everywhere. $\square$

**Corollary 1.** *Under assumption (B8), the bound can be expressed in terms of the finite error $\varepsilon_\pi$:*

$$I(\mathbf{z}_0; \hat{\mathbf{x}}) \;\geq\; \mathcal{L}_I \;-\; \varepsilon_\pi. \tag{13}$$

**Remark 1** (Gaussian Note). *For a standard normal prior $r(\mathbf{z}_0) = \mathcal{N}(\mathbf{0}, \mathbf{I})$, the term $-\log r(\mathbf{z}_0)$ becomes $\frac{1}{2}\|\mathbf{z}_0\|^2 + \text{const}$. Thus, the tractable objective $\mathcal{L}_I$ is equivalent to maximizing $\mathbb{E}_{q_\phi(\mathbf{z}_0, \hat{\mathbf{x}})}[\log Q_\xi(\mathbf{z}_0 \mid \hat{\mathbf{x}})] + \frac{1}{2}\mathbb{E}_{q_\phi(\mathbf{z}_0)}[\|\mathbf{z}_0\|^2]$ up to an additive constant.*

### A.3 PROOF OF LEMMA 1 (UPPER BOUND ON INFORMATION LEAKAGE)

The proof of Lemma 1 establishes a bound on the mutual information $I(y; \mathbf{z}_0) = \mathbb{E}_{\mathbf{z}_0 \sim q_\phi(\mathbf{z}_0)}[D_{KL}(p(y \mid \mathbf{z}_0)\|p(y))]$ (Cover & Thomas, 2006). The derivation proceeds by first establishing a pointwise bound for a single realization of $\mathbf{z}_0$ and then averaging this bound over the distribution $q_\phi(\mathbf{z}_0)$.

#### A.3.1 DERIVATION OF THE MAIN BOUND

The proof leverages a standard chain of inequalities relating various $f$-divergences, including the inequality $D_{KL}(P\|R) \leq \frac{1}{p_{\min}}\|P - R\|_1^2$ for distributions with minimum probability $p_{\min}$ (Sason, 2015, Eq. 10), and Pinsker's inequality, $\|P - R\|_1^2 \leq 2D_{KL}(P\|R)$ (Csiszár & Körner, 2011).

Let $P := p(y \mid \mathbf{z}_0)$ be the true posterior, $R := p(y)$ be the class prior, and $Q := q_\omega(y|\mathbf{z}_0)$ be the discriminator's predictive distribution. The derivation begins by relating the KL divergence to the squared $\ell_1$ norm:

$$D_{KL}(P\|R) \leq \frac{1}{p_{\min}}\|P - R\|_1^2. \tag{14}$$

We bound the $\ell_1$ norm by introducing the discriminator's prediction $Q$ as an intermediate point. By the triangle inequality and the property that $(a + b)^2 \leq 2(a^2 + b^2)$, we have:

$$\|P - R\|_1^2 \leq (\|P - Q\|_1 + \|Q - R\|_1)^2 \leq 2\left(\|P - Q\|_1^2 + \|Q - R\|_1^2\right). \tag{15}$$

Applying Pinsker's inequality to both terms on the right-hand side yields:

$$\|P - R\|_1^2 \leq 2\left(2D_{KL}(P\|Q) + 2D_{KL}(Q\|R)\right) = 4\left(D_{KL}(P\|Q) + D_{KL}(Q\|R)\right). \tag{16}$$

Combining these results gives the complete pointwise bound for a given $\mathbf{z}_0$:

$$D_{KL}(P\|R) \leq \frac{4}{p_{\min}} \left(D_{KL}(P\|Q) + D_{KL}(Q\|R)\right). \tag{17}$$

To finalize the proof, we take the expectation of both sides with respect to $\mathbf{z}_0 \sim q_\phi(\mathbf{z}_0)$. By the linearity of expectation, this yields:

$$I(y;\mathbf{z}_0) = \mathbb{E}_{\mathbf{z}_0}[D_{KL}(P\|R)] \leq \frac{4}{p_{\min}} \left(\mathbb{E}_{\mathbf{z}_0}[D_{KL}(P\|Q)] + \mathbb{E}_{\mathbf{z}_0}[D_{KL}(Q\|R)]\right). \tag{18}$$

The two terms on the right are precisely the quantities bounded by our finite-error assumptions (B1) and (B2). Substituting the bounds $\varepsilon_D$ and $\eta$ completes the proof:

$$I(y;\mathbf{z}_0) \leq \frac{4}{p_{\min}}(\varepsilon_D + \eta). \tag{19}$$

### A.3.2 DERIVATION FOR THE UNIFORM-TARGET VARIANT

The proof for the case where the encoder's target distribution is the uniform distribution $U$ follows a similar structure. The encoder's objective is now bounded by $\mathbb{E}_{\mathbf{z}_0}[D_{KL}(Q\|U)] \leq \eta'$. We introduce $U$ as a second intermediate distribution. Applying the triangle inequality iteratively gives $\|P - R\|_1 \leq \|P - Q\|_1 + \|Q - U\|_1 + \|U - R\|_1$. Using the inequality $(a + b + c)^2 \leq 3(a^2 + b^2 + c^2)$ and applying Pinsker's to each term yields:

$$\begin{aligned}
\|P - R\|_1^2 &\leq 3\left(\|P - Q\|_1^2 + \|Q - U\|_1^2 + \|U - R\|_1^2\right) \\
&\leq 6\left(D_{KL}(P\|Q) + D_{KL}(Q\|U) + D_{KL}(R\|U)\right).
\end{aligned} \tag{20}$$

Taking the expectation over $\mathbf{z}_0$ and noting that $D_{KL}(R\|U) = D_{KL}(p(y)\|U)$ is a constant, we substitute the finite-error bounds to arrive at the final result:

$$I(y;\mathbf{z}_0) \leq \frac{6}{p_{\min}}(\varepsilon_D + \eta' + D_{KL}(p(y)\|U)). \tag{21}$$

### A.4 PROOF OF LEMMA 2 (LOWER BOUND ON CONDITIONAL RESIDUAL INFORMATION)

The proof is constructed in three stages. First, we establish a lower bound on the total information the full latent code $\mathbf{z}$ must contain about the input $\mathbf{x}$. Second, we decompose this total information into components corresponding to the two subspaces. Finally, we derive an upper bound on the information capacity of the task-relevant subspace $\mathbf{z}_1$.

### A.4.1 STEP 1: RATE-DISTORTION LOWER BOUND ON TOTAL INFORMATION

Our technical condition (T2) establishes the Markov chain $\mathbf{x} \to \mathbf{z} \to \hat{\mathbf{x}}$. By the Data Processing Inequality, the mutual information cannot increase along this chain (Cover & Thomas, 2006), which implies:

$$I(\mathbf{x};\mathbf{z}) \geq I(\mathbf{x};\hat{\mathbf{x}}). \tag{22}$$

The Shannon rate-distortion function $R_X(D)$ defines the minimum possible rate $I(\mathbf{x};\hat{\mathbf{x}})$ required to achieve an average distortion of at most $D$ (Berger, 1971). Our finite-error assumption (B3) states that our model achieves a reconstruction error bounded by $\delta_{rec}$. Therefore, the rate achieved by our model must be at least the theoretical minimum:

$$I(\mathbf{x};\hat{\mathbf{x}}) \geq R_X(\delta_{rec}). \tag{23}$$

Combining these inequalities yields the lower bound on the total information encoded in the latent space:

$$I(\mathbf{x};\mathbf{z}) \geq R_X(\delta_{rec}). \tag{24}$$

### A.4.2 STEP 2: DECOMPOSING INFORMATION VIA THE CHAIN RULE

We apply the chain rule of mutual information (Cover & Thomas, 2006) to decompose the total information in the composite latent variable $\mathbf{z} = (\mathbf{z}_1, \mathbf{z}_0)$:

$$I(\mathbf{x};\mathbf{z}) = I(\mathbf{x};\mathbf{z}_1) + I(\mathbf{x};\mathbf{z}_0 \mid \mathbf{z}_1). \tag{25}$$

Rearranging and substituting the result from Step 1 gives a lower bound on the information uniquely available in the residual subspace:

$$I(\mathbf{x};\mathbf{z}_0 \mid \mathbf{z}_1) \geq R_X(\delta_{rec}) - I(\mathbf{x};\mathbf{z}_1). \tag{26}$$

### A.4.3 STEP 3: DERIVING THE UPPER BOUND ON TASK-RELEVANT INFORMATION

The final step is to derive an upper bound on $I(\mathbf{x}; \mathbf{z}_1)$. We begin with the definition of mutual information, $I(\mathbf{x}; \mathbf{z}_1) = h(\mathbf{z}_1) - h(\mathbf{z}_1 \mid \mathbf{x})$, where $h(\cdot)$ is the differential entropy.

From our stochastic encoder assumption (T1), the conditional entropy $h(\mathbf{z}_1 \mid \mathbf{x})$ is simply the entropy of the $d_1$-dimensional Gaussian noise component, a standard result:

$$h(\mathbf{z}_1 \mid \mathbf{x}) = \frac{d_1}{2} \log(2\pi e \sigma^2). \tag{27}$$

To bound the marginal entropy $h(\mathbf{z}_1)$, we first relate it to the conditional entropy $h(\mathbf{z}_1 \mid y)$ via the identity

$$h(\mathbf{z}_1) = h(\mathbf{z}_1 \mid y) + I(\mathbf{z}_1; y) \ \text{ with } \ I(\mathbf{z}_1; y) \le H(y), \tag{28}$$

which implies

$$h(\mathbf{z}_1) \le H(y) + h(\mathbf{z}_1 \mid y). \tag{29}$$

The conditional entropy is maximized by a Gaussian distribution (Cover & Thomas, 2006), and by applying Jensen's inequality, we can bound it by the average trace of the per-class covariance matrices, $\Sigma_y$:

$$h(\mathbf{z}_1 \mid y) = \mathbb{E}_y[h(\mathbf{z}_1 \mid Y = y)] \le \frac{d_1}{2} \log\left(2\pi e \frac{\mathbb{E}_y[\mathrm{tr}(\Sigma_y)]}{d_1}\right). \tag{30}$$

(Boyd & Vandenberghe, 2004). The structural loss assumption (B4) provides a bound on this average trace. Let $\mu_y = \mathbb{E}_{\mathbf{x}|y}[\mathbf{z}_1]$ be the true conditional mean. The expected loss for class $y$ can be decomposed as

$$\mathbb{E}_{\mathbf{x}|y}\big[\|\mathbf{z}_1 - \mathbf{M}_y\|_2^2\big] = \mathrm{tr}(\Sigma_y) + \|\mu_y - \mathbf{M}_y\|_2^2. \tag{31}$$

Since $\|\mu_y - \mathbf{M}_y\|_2^2 \ge 0$, taking the expectation over $y$ gives $\delta_\tau \ge \mathbb{E}_y[\mathrm{tr}(\Sigma_y)]$. Substituting this into the entropy bound yields:

$$h(\mathbf{z}_1 \mid y) \le \frac{d_1}{2} \log\left(2\pi e \frac{\delta_\tau}{d_1}\right). \tag{32}$$

Assembling these results provides the upper bound for the marginal entropy:

$$h(\mathbf{z}_1) \le H(y) + \frac{d_1}{2} \log\left(2\pi e \frac{\delta_\tau}{d_1}\right). \tag{33}$$

Finally, substituting the exact expression for $h(\mathbf{z}_1 \mid \mathbf{x})$ and the upper bound for $h(\mathbf{z}_1)$ into the mutual information identity gives:

$$I(\mathbf{x}; \mathbf{z}_1) \le \left[H(y) + \frac{d_1}{2} \log\left(2\pi e \frac{\delta_\tau}{d_1}\right)\right] - \left[\frac{d_1}{2} \log(2\pi e \sigma^2)\right] = H(y) + \frac{d_1}{2} \log\left(\frac{\delta_\tau}{d_1 \sigma^2}\right). \tag{34}$$

Substituting this upper bound for the subtracted term in the inequality from Step 2 yields the final result of the lemma:

$$\boxed{I(\mathbf{x}; \mathbf{z}_0 \mid \mathbf{z}_1) \ge R_X(\delta_{rec}) - H(y) - \frac{d_1}{2} \log\left(\frac{\delta_\tau}{d_1 \sigma^2}\right)} \tag{35}$$

## B ARCHITECTURES AND HYPERPARAMETERS

All models were implemented in PyTorch and trained using the PyTorch Lightning framework. We provide a detailed breakdown of the architectures and hyperparameters used for the Morpho-MNIST and CelebA experiments. All activation functions, unless otherwise specified, are LeakyReLU.

### B.1 MORPHO-MNIST ARCHITECTURE

For the ablation study and Lemma 2 validation on Morpho-MNIST, we used a convolutional architecture with a fully-connected bottleneck. The concept-extraction module consists of the Encoder ($E_\phi$) and Generator ($G_\theta$), which is complemented by three auxiliary networks from the adversarial module: a Classifier ($C_\psi$), a multi-task Latent-Residual Discriminator ($D_{R,\omega}$), and a main image Discriminator ($D_{x,\delta}$) which also contains the Q-network head. The specific layer configurations for each component are detailed in Table 3.

Table 3: Detailed architecture for Morpho-MNIST experiments. KS denotes kernel size, S denotes stride, and P denotes padding. The output shape is shown for a batch size of $N$.

| Component | Layer Type | Output Shape |
|---|---|---|
| **Encoder** ($E_\phi$) | | |
| | Conv2d (KS 3, S 1, P 1), LeakyReLU | $N \times 32 \times 28 \times 28$ |
| | MaxPool2d (KS 2, S 2) | $N \times 32 \times 14 \times 14$ |
| | Conv2d (KS 3, S 1, P 1), LeakyReLU | $N \times 64 \times 14 \times 14$ |
| | MaxPool2d (KS 2, S 2) | $N \times 64 \times 7 \times 7$ |
| | Conv2d (KS 3, S 1, P 1), LeakyReLU | $N \times 64 \times 7 \times 7$ |
| | Flatten | $N \times 3136$ |
| | Linear, LeakyReLU | $N \times 128$ |
| | Linear (Output $\mathbf{z}$) | $N \times 16$ |
| **Generator** ($G_\theta$) | | |
| | Linear (Input $\mathbf{z}$), LeakyReLU | $N \times 128$ |
| | Linear, LeakyReLU | $N \times 3136$ |
| | Unflatten | $N \times 64 \times 7 \times 7$ |
| | Conv2d (KS 3, S 1, P 1), LeakyReLU | $N \times 64 \times 7 \times 7$ |
| | Conv2d (KS 1, S 1), PixelShuffle(2), LeakyReLU | $N \times 64 \times 14 \times 14$ |
| | Conv2d (KS 3, S 1, P 1), LeakyReLU | $N \times 32 \times 14 \times 14$ |
| | Conv2d (KS 1, S 1), PixelShuffle(2), LeakyReLU | $N \times 32 \times 28 \times 28$ |
| | Conv2d (KS 3, S 1, P 1), Tanh | $N \times 1 \times 28 \times 28$ |
| **Classifier** ($C_\psi$) | | |
| | Linear (Input $\mathbf{z}_1$), LeakyReLU | $N \times 64$ |
| | Linear | $N \times 10$ |
| **Latent-Residual Discriminator** ($D_{R,\omega}$) | | |
| | Linear (Input $\mathbf{z}_0$), LeakyReLU | $N \times 64$ |
| | Head 1: Linear (Classification, $\mathcal{L}_{D,R}$) | $N \times 10$ |
| | Head 2: Linear (Prior Matching, $\mathcal{L}_{D,\pi}$) | $N \times 1$ |
| **Image Discriminator** ($D_{x,\delta}$) **& Q-Network** ($Q_\xi$) | | |
| | Conv2d (KS 3, S 1, P 1), LeakyReLU | $N \times 32 \times 28 \times 28$ |
| | MaxPool2d (KS 2, S 2) | $N \times 32 \times 14 \times 14$ |
| | Conv2d (KS 3, S 1, P 1), LeakyReLU | $N \times 64 \times 14 \times 14$ |
| | MaxPool2d (KS 2, S 2) | $N \times 64 \times 7 \times 7$ |
| | Conv2d (KS 3, S 1, P 1), LeakyReLU | $N \times 64 \times 7 \times 7$ |
| | AdaptiveAvgPool2d | $N \times 64 \times 1 \times 1$ |
| | Flatten | $N \times 64$ |
| | Head 1: Linear (Real/Fake, $\mathcal{L}_{D,x}$) | $N \times 1$ |
| | Head 2: Linear (Q Head $\mu$, $\mathcal{L}_Q$) | $N \times 8$ |
| | Head 3: Linear (Q Head $\log \sigma^2$, $\mathcal{L}_Q$) | $N \times 8$ |

## B.2 CELEBA ARCHITECTURE

For the higher-resolution CelebA dataset, we employed a deeper, fully-convolutional network (FCN) architecture with residual blocks (He et al., 2016). The Encoder consists of 5 residual downsampling blocks that reduce the spatial resolution from $128 \times 128$ to $4 \times 4$. This produces a spatial latent map, where the latent code $\mathbf{z}$ is a tensor of shape $16 \times 4 \times 4$. The partition into $\mathbf{z}_1$ and $\mathbf{z}_0$ is performed along the channel dimension; we designate the first 4 channels as the task-relevant subspace $\mathbf{z}_1$ (a $4 \times 4 \times 4$ tensor) and the remaining 12 channels as the residual subspace $\mathbf{z}_0$ (a $12 \times 4 \times 4$ tensor). The Generator mirrors this structure with 5 residual upsampling blocks. The Image Discriminator ($D_{x,\delta}$) architecture was similarly scaled up to match the generator's depth and complexity.

### B.3 HYPERPARAMETERS

All models were trained for 100 epochs with a batch size of 256 for Morpho-MNIST and 64 for CelebA. We used the AdamW optimizer (Loshchilov & Hutter, 2019) with $\beta_1 = 0.5$, $\beta_2 = 0.999$, and a weight decay of $10^{-4}$. The learning rate for the concept-extraction module ($\Theta_M$) was set to $3 \times 10^{-4}$, while the adversarial module ($\Theta_A$) used a lower learning rate of $3 \times 10^{-6}$. For the Morpho-MNIST experiments, the reconstruction loss $\mathcal{L}_{\hat{x}}$ was mean squared error (MSE). For CelebA, we used a multi-layer VGG-19 perceptual loss (Johnson et al., 2016). The weights for each loss component in the full PRISM model are listed in Table 4.

Table 4: Loss component weights for the full PRISM model on Morpho-MNIST.

| Hyperparameter | Value |
|---|---|
| Reconstruction ($\gamma_{\hat{x}}$) | 1.0 |
| Image Adversary ($\gamma_x$) | 1.0 |
| Classification ($\gamma_y$) | 1.0 |
| Structural ($\gamma_\tau$) | 0.01 |
| Latent Adversary ($\gamma_L$) | 0.1 |
| InfoGAN ($\gamma_I$) | 0.01 |
| Prior Matching ($\gamma_\pi$) | 0.01 |
| R1 Gradient Penalty ($\gamma_{R1}$) | 10.0 |

## C MORPHO-MNIST QUALITATIVE RESULTS

To complement the quantitative analysis in the main text, we provide qualitative visualizations on the Morpho-MNIST dataset that offer an intuitive validation of the latent structure learned by our model (Figures 4 to 7. Comparing the full PRISM model to key ablated configurations, we probe the representations through: (i) t-SNE projections (Maaten & Hinton, 2008) to visualize class-based clustering, (ii) latent code swapping to test the separation of content and style, (iii) traversals to inspect the learned factors of variation, and (iv) mean code replacement to verify the distinct semantic role of each subspace.

## D LLM USAGE

We used ChatGPT (OpenAI) and Gemini 2.5 Pro (Google) solely for copy-editing (grammar, phrasing, and clarity) and minor LaTeX formatting suggestions. No ideas, analyses, datasets, code, figures, or results were generated by these tools. All edits were reviewed and verified by the authors, who take full responsibility for the content.

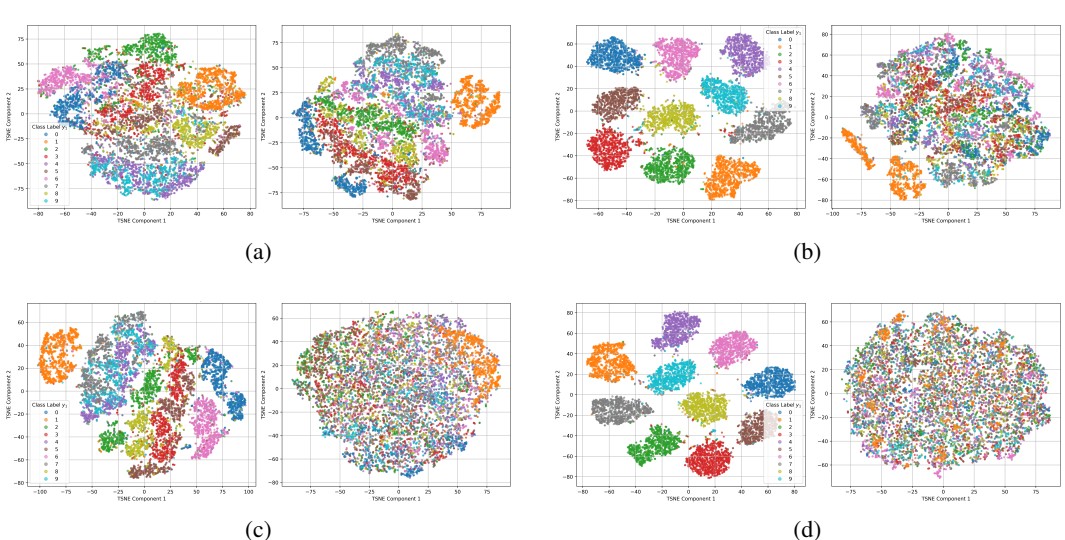

Figure 4: **t-SNE visualization of latent subspace gemoetry.** Each panel compares the structure of the concept-relevant subspace $\mathbf{z}_1$ (left) and the residual subspace $\mathbf{z}_0$ (right). (a) Baseline $(\mathcal{L}_{\hat{x}}, \mathcal{L}_{G,x})$, (b) Baseline with classification and structural losses $(\mathcal{L}_y, \mathcal{L}_\tau)$, (c) Baseline with residual subspace regularizers $(\mathcal{L}_{G,R}, \mathcal{L}_I, \mathcal{L}_{G,\pi})$, (d) the full PRISM model. Points are colored by their ground-truth class label, visually revealing the degree to which the class identity is either isolated in $\mathbf{z}_1$ or leaks into $\mathbf{z}_0$. The comparison clearly illustrates the intended effect of each component: the losses on $\mathbf{z}_1$ induce a distinct, class-separated geometry, while the regularizers on $\mathbf{z}_0$ actively purge class-specific information, resulting in a homogeneous, unstructured distribution.

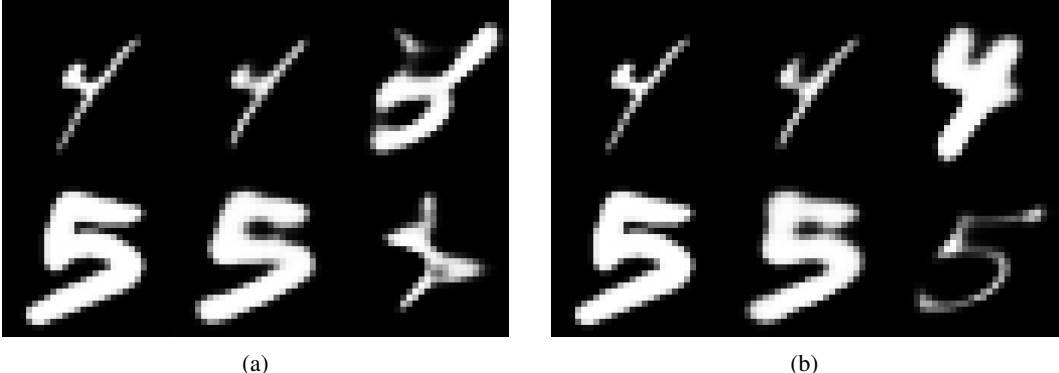

Figure 5: **Testing factorization via residual subspace $\mathbf{z}_0$ swapping.** We compare the factorization performance of (a) the Baseline model $(\mathcal{L}_{\hat{x}}, \mathcal{L}_{G,x})$ and (b) the full PRISM model. For each pair of source images $\mathbf{x}$ (left column), we show their standard reconstructions $\hat{\mathbf{x}}$ (middle column) and the result of swapping their residual codes $\mathbf{z}_0$ while preserving their original concept codes $\mathbf{z}_1$ (right column). A successful factorization should combine the digit's identity, encoded in $\mathbf{z}_1$, with stylistic attributes - such as thickness and angle - from the other image's $\mathbf{z}_0$. The full model successfully achieves this attribute transfer, whereas the baseline fails to preserve the original digit identity, indicating a critical leakage of class information into its residual subspace.

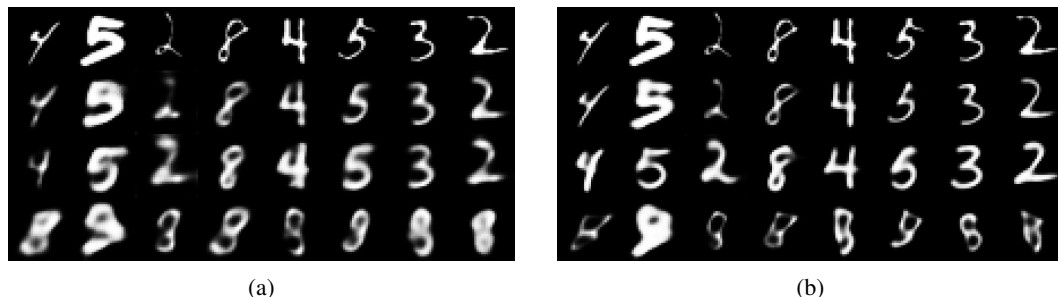

(a)  (b)

Figure 6: **Verifying the semantic role of the structural loss $\mathcal{L}_\tau$.** This experiment isolates the contribution of the structural loss $\mathcal{L}_\tau$ by comparing (a) a model trained without it against (b) the full PRISM model. Each of the eight columns displays a different source sample. The four rows depict: (1) the original image $\mathbf{x}$; (2) its standard reconstruction $\hat{\mathbf{x}}$; (3) reconstruction using the sample's concept code $\mathbf{z}_1$ but the dataset-average residual code $\mathbf{z}_0$; and (4) reconstruction using the sample's $\mathbf{z}_0$ but the dataset-average $\mathbf{z}_1$. The key comparison is in row (3): for the full model (b), replacing the residual code with its mean effectively normalizes all stylistic variation, leaving a canonical rendering of the digit's identity. In contrast, the model lacking $\mathcal{L}_\tau$ (a) fails to do so, as stylistic information clearly persists in its reconstructions. This directly demonstrates that $\mathcal{L}_\tau$ is essential for compacting the intra-class variation within the concept subspace $\mathbf{z}_1$, forcing it to encode a style-invariant representation of the class.

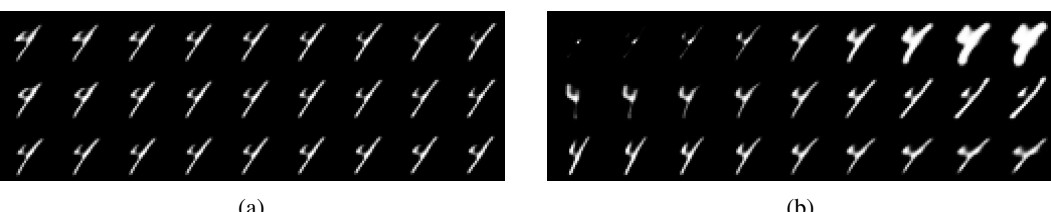

(a)  (b)

Figure 7: **Inspecting learned factors of variation via latent traversal of $\mathbf{z}_0$.** This experiment demonstrates the necessity of the mutual information objective ($\mathcal{L}_I$) for preventing degenerate solutions in the residual subspace. We compare (a) a Baseline model ($\mathcal{L}_{\hat{x}}$, $\mathcal{L}_{G,x}$) equipped only with a latent-residual adversary ($\mathcal{L}_{G,R}$) against (b) the full PRISM model. Each of the three rows corresponds to a traversal along one of the top three principal components of the aggregated posterior of $\mathbf{z}_0$, from $-2$ to $+2$ standard deviations. The comparison reveals precisely why this objective is necessary: relying exclusively on the latent-residual adversary, as in model (a), leads to a degenerate solution where the generator learns to simply discard the nuisance information from $\mathbf{z}_0$, causing the representation to collapse. This failure is prevented in the full model (b), where the InfoGAN-style loss ensures the subspace remains informative and encodes salient, class-agnostic attributes.

