# OpenReview forum: "PRISM: A Principled Framework for Supervised Disentanglement via Bipartite Factorization"
_ICLR.cc/2026/Conference — Submitted to ICLR 2026_

### Official Review · Reviewer_MCZj · 2025-10-24

**Soundness:** 4
**Presentation:** 4
**Contribution:** 3
**Rating:** 6
**Confidence:** 3

**Summary:**

In this work the authors construct set of neural networks and losses to distill two aspects of the latent code of datapoints given a dataset of labels: those that impact the label and those that are independent of it. They demonstrate the effectiveness of the extraction procedure, perform an ablation study, provide supporting theory.

**Strengths:**

- The paper is very clear, from problem statement, through literature review, to theory and experiments
- The contribution is novel, elaborate, and appears to work
- The authors perform a meaningful ablation study

Broadly, it just seemed relatively well executed.

I liked the paper, and think it makes a good contribution to ICLR as is. I oscillated between weak acceptance and acceptance. My (minor) negative comments are included in the weaknesses section. Broadly, if the authors could do more to convince me that their method was simply better than alternatives, and if I was certain there were no other more recent models (Fader is 2017) [I don't know this literature well, I will defer to other reviewers on this], I would lean more towards acceptance.

**Weaknesses:**

- Personally, I did not find the maths added very much. The final predictions they ended up with felt relatively intuitive - (i) more reconstruction requires more info in z_0, (ii) less dimensionality of z_1 pushes more info into z_0, (iii) removing non-class info from z_1 pushes it into z_0. It is indeed nice to tie these intuitions to a principled framework with attached assumptions etc. but I didn't get a lot scientifically from it. I'd be interested to hear what the authors think it taught them.

- Could figure 1, especially the text, be made bigger.

- Appendix C seemed very important, personally I would prioritise pushing that into the main paper, at the expense of, for example, section 5.3, section 4, and perhaps some of section 2. Further, is FADER this the only existing comparable method? And the improvements the method achieves over FADER are minimal - the two label leakages are within error bars of one another. Hence, it doesn't seem that the method is necessarily better, just different, muddying the important contribution of this paper?

**Questions:**

See above.

---

> ### Author Response · Authors · 2025-12-01
>
> We sincerely thank you for your time and for your insightful review. We appreciate your recognition of our work's novelty and the paper's clarity. We are particularly grateful for your thought-provoking questions regarding the practical utility of our theoretical analysis and the qualitative advantages of our method over alternatives. We also deeply value your constructive advice on the paper's organization; we have restructured the manuscript accordingly to improve the presentation. Below, we address your questions in detail.
>
> ### Reviewer comment
>
> > "Personally, I did not find the maths added very much. The final predictions they ended up with felt relatively intuitive... I'd be interested to hear what the authors think it taught them."
>
> **Response:** We thank the reviewer for this excellent question, as it gets to the heart of our paper's contribution. We wish to clarify that the primary utility of our theoretical framework (Section 4) was not to be purely descriptive, but to serve as a prescriptive guide for our architectural design and experimental validation. It allowed us to move from an ad-hoc collection of losses to a principled system where the role of each component is understood.
>
> For example, the analysis in **Section 4.3**, which is formalized in **Lemma 2**, is what allowed us to formally identify the three key "architectural levers" that govern the information partition: (i) reconstruction fidelity $\delta\_{\mathrm{rec}}$, (ii) concept subspace dimensionality $d\_1$, and (iii) intra-class compactness $\delta\_{\tau}$. This theoretical foundation is what directly *enabled* us to design the targeted validation experiments in **Section 5.3 (Figure 2)**, which empirically confirm the causal links predicted by our theory. This deeper understanding also had direct, practical benefits, providing invaluable guidance for our hyperparameter search. By understanding the theoretical relationship between, for example, the structural loss weight $\gamma\_{\tau}$ (which controls $\delta\_{\tau}$) and the information partition, we could form strong, educated hypotheses about the relative importance of terms.
>
> Furthermore, the analysis informed critical design choices by revealing non-obvious practical constraints. A key example of this is the analysis that informed our choice of the adversarial target distribution. From a practical standpoint, a simple **uniform distribution ($U$)** is an appealing target for the adversarial classifier. It is easy to implement and does not require estimating the dataset's empirical class prior $p(y)$. One might intuitively assume this would work well.
>
> However, our formal analysis for **Lemma 1** revealed a *provable flaw* in this intuitive choice. When we derived the bound for this "simpler" uniform-target variant (in **Appendix A.3.2**), the resulting bound on information leakage $I(y; z\_0)$ became significantly *looser*. Specifically, the bound contains an **additional positive term: $D\_{\mathrm{KL}}(p(y)\||U)$**.
>
> This term is the KL divergence between the dataset's *true* class distribution $p(y)$ and the uniform distribution $U$. This divergence is zero *only if* the dataset is perfectly class-balanced. For any realistic, *imbalanced* dataset, this term becomes a large positive constant, which weakens the theoretical guarantee of information suppression. This analysis directly justified our nuanced approach: for the imbalanced CelebA dataset, we used the empirical prior $p(y)$ as the target, which, while slightly more complex, is provably more robust. Conversely, this theory also explains why the simpler uniform target was sufficient for our Morpho-MNIST experiments, where the dataset is balanced and this problematic $D\_{\mathrm{KL}}(p(y)\||U)$ term is negligible.

---

> ### Author Response · Authors · 2025-12-01
>
> ### Reviewer comment
>
> > "Appendix C seemed very important, personally I would prioritise pushing that into the main paper, at the expense of, for example, section 5.3, section 4, and perhaps some of section 2. Further, is FADER this the only existing comparable method?"
>
> **Response:** We thank the reviewer for this excellent suggestion and for highlighting the importance of this comparison. We completely agree that the quantitative results against Fader Networks are a critical piece of our evaluation. Following the reviewer's advice, we have moved this analysis from the appendix directly into the main paper. It is now presented in **Section 5.4 and Table 2** of the revised manuscript, where we hope it more clearly contextualizes our performance.
>
> Regarding the choice of baseline, we selected Fader Networks (Lample et al., 2017) as our primary point of comparison for principled reasons. It is a seminal and highly influential work that established the core adversarial approach for the specific problem of supervised, bipartite factorization. This makes it an ideal and direct benchmark for our goal, allowing for a clear, apples-to-apples comparison of how effectively a known, supervised attribute can be isolated from all other sources of variation.
>
> While our literature review in **Section 2** situates PRISM within the broader landscape of disentanglement research - including unsupervised (lines 80-89) and weakly-supervised (lines 97-107) approaches - these methods often have different architectural assumptions or scientific goals (e.g., full disentanglement from structural signals vs. bipartite factorization from a semantic label), making a direct quantitative comparison less informative. Therefore, we believe the focused comparison with Fader Networks provides the clearest and most scientifically rigorous assessment of our specific contribution. We thank the reviewer again for helping us improve the presentation and impact of our results.
>
> ### Reviewer comment
>
> > "...the improvements the method achieves over FADER are minimal - the two label leakages are within error bars of one another. Hence, it doesn't seem that the method is necessarily better, just different, muddying the important contribution of this paper?"
>
> **Response:** We thank the reviewer for this sharp observation, which allows us to clarify what we see as the most significant contribution of our work. While the direct label leakage metric is indeed comparable, we believe this similarity obscures a fundamental architectural difference that makes our approach not just different, but qualitatively more powerful.
>
> Fader Networks operates by making its latent space invariant to the supervised attribute, which is then reintroduced to the generator via a fixed, one-hot class label. These labels are geometrically arbitrary; the representation for 'digit 1' is no closer to 'digit 7' than it is to 'digit 8'. In contrast, PRISM does not discard and reintroduce the attribute. Instead, our encoder learns to **distill** the core essence of the concept from the input image into a continuous, semantically rich subspace, $z\_1$.
>
> The key advantage of this approach is that the geometry of this learned subspace becomes meaningful. The proximity between class clusters in $z\_1$ comes to reflect their actual visual similarity, creating a far more powerful, purpose-built feature embedding. For downstream tasks, a representation that understands the nuanced relationships *between* classes is potentially more robust and generalizable than one that has been explicitly trained to be class-invariant.
>
> Furthermore, this continuous concept subspace becomes an invaluable tool for model interpretability. It allows us to move beyond simply labeling an image and begin to ask what the model has learned about the concept itself. For instance, as shown in our qualitative experiments in **Figure 3**, we can generate a "canonical" version of a concept by averaging its $z\_1$ embeddings, effectively normalizing all stylistic variation. This provides a window into the model's core understanding of a class - an analysis that is simply not possible within the Fader Networks framework.
>
> ### Reviewer comment
>
> > "Could figure 1, especially the text, be made bigger."
>
> **Response:** We thank the reviewer for this practical feedback. We agree that the text in Figure 1 was too small for easy readability. We have revised the figure in the new version of the manuscript, increasing the font size and improving the layout to ensure all components and labels are clear and legible. We appreciate the reviewer pointing this out and helping us improve the paper's presentation.

---

### Official Review · Reviewer_4kLh · 2025-10-31

**Soundness:** 2
**Presentation:** 1
**Contribution:** 1
**Rating:** 2
**Confidence:** 4

**Summary:**

This paper proposes supervised disentangled representation learning by partitioning the latent space into two codes: one that captures the supervised factor of variation and the other that captures the remaining (class-agnostic) information. To this end, the authors design an adversarial training framework that encourages the first latent to be class-relevant while constraining the second to be class-irrelevant. The proposed method is validated on Morpho-MNIST via component-wise ablations and by empirically estimating a lower bound on the conditional residual information which indicates low leakage of supervised information into the residual latent code.

**Strengths:**

- This paper addresses an important challenge in disentangled representation learning under weak supervision.
- It provides an information-theoretic analysis with quantitative bounds (e.g., on information leakage and conditional residual information).

**Weaknesses:**

- The overall presentation is not well organized, which makes it difficult to understand the paper’s main claim and method. The approach is introduced with many equations and components without high-level intuition or justification. (Please see the questions below.)
- The proposed method introduces too many objectives (at least seven within an adversarial scheme), which overly complicates the framework. It makes hyperparameter search very complex (the method has at least seven loss-weighting coefficients such as
 $\lambda_x, \lambda_y, etc$.
- A major concern is the lack of comparisons to prior work (e.g., [1]) and the absence of evaluation on common disentanglement metrics (e.g., DCI, FactorVAE score, MIG as in [1]), which makes it hard to assess the method’s effectiveness relative to prior work. Instead, the paper reports probe-gap (without reference or justification) and ARI, which appears orthogonal to disentanglement quality. Moreover, the evaluation is limited to a single, simple synthetic dataset (Morpho-MNIST).

**Reference**

[1] Locatello et al., Weakly-supervised disentanglement without compromises, in ICML 2020.

**Questions:**

- Why is the structural loss (L196) necessary for disentanglement? Isn’t this objective orthogonal to disentanglement of the representation?
- Why do we have to maximize $I(z_0;\hat {x})$? Doesn’t the reconstruction loss already encourage $z_0$ to contain information for reconstructing $x$?
- What is $R_X(\sigma_{rec})$ in L313 and how is it measured?
- How did the authors select and tune the numerous hyperparameters for the loss-weighting coefficients?
- Section 3.1 begins by describing the architectures and optimization framework without any overview or justification, which makes it hard for readers to follow the overall flow. From the beginning, it is not even clear why the proposed method requires an adversarial framework. Providing a brief overview would improve clarity.
- Similarly, Section 3.2 lists many components without sufficient description. For example, it starts by introducing the loss $L_M$​, which consists of seven loss terms, but readers have no context for each term. Although the following sentences group these terms into three categories, the section still lacks justifications or high-level intuitions for the losses—e.g., why the concept-relevant subspace must be structured, why the residual subspace should be shaped, and what role each loss term plays.
- Please add equation numbering.
- Please consider redrawing Figure 1 for readability; the text in the figure is too small to read and the figure looks too complex.

---

> ### Author Response · Authors · 2025-12-01
>
> Thank you for taking the time to prepare your review.
>
> The extensive material that we have provided, both in the paper’s main body and in the Appendices (concept description, formal proofs, evaluation results), could probably be organized somewhat better to facilitate its careful absorption, as the majority of concerns that you raised in your review were addressed or explained throughout the original paper.
>
> We find the two (of the three) objections, identified as key weaknesses of the paper, to be invalid. Regarding your first concern (too many objectives), we explicitly demonstrated through the ablation study that all components employed for learning the disentangled representation were necessary, and that each played a clear and intuitive role (as presented in Section 5.2 and Appendix D of the original submission). Regarding your second objection (a lack of comparison to prior work and quantitative disentanglement metrics), we had quantitatively compared the performance of the proposed method against the adequate reference approach (Appendix C of the original submission). We also provided disentanglement metrics, such as DCI and the SAP score (Appendix C of the original submission). Moreover, our method was verified not only on the Morpho-MNIST dataset, as you mentioned, but also on a much more demanding, real-world Celeb-A image dataset (Section 5.4 of the original submission), confirming the effectiveness of the proposed approach.
>
> We agree that the presentation could be improved (the third key weakness), and to increase its clarity, we have extended the paper to explain better the concept of a principled, multi-objective approach, where each loss component used in design is introduced to solve a specific, predictable failure mode of a simpler baseline. Additionally, to address some of your objections, we have moved the contents, which were apparently difficult to find, from the appendices to the main body of the paper. Below, please find a list of our detailed answers to your specific concerns.
>
> ### Reviewer comment
>
> > "The overall presentation is not well organized, which makes it difficult to understand the paper’s main claim and method. The approach is introduced with many equations and components without high-level intuition or justification."
>
> > **Question 5:** "Section 3.1 begins by describing the architectures and optimization framework without any overview or justification, which makes it hard for readers to follow the overall flow..."
>
> > **Question 6:** "Similarly, Section 3.2 lists many components without sufficient description... the section still lacks justifications or high-level intuitions for the losses..."
>
> **Response:** We thank the reviewer for the feedback on the paper's presentation. We agree that a consolidated overview of the framework's logic at the beginning of Section 3 would improve the manuscript's clarity. We have therefore incorporated this suggestion directly into the revised paper.
>
> For the reviewer's convenience, we will briefly summarize the core logic that is now presented upfront in the revised **Section 3**. This overview connects the high-level motivation to the specific analyses and results included throughout the paper.
>
> Our central mechanism, as motivated in our theoretical analysis (**Section 4.3**), is to create a deliberate conflict that the optimization must resolve. The framework renders the concept-relevant subspace $\mathbf{z}\_1$ an inefficient channel for non-class variation by imposing an information bottleneck. This is achieved via the structural prototype loss ($\mathcal{L}\_{\tau}$), and its role in creating a compact concept space is empirically validated in our ablation study (**Table 1**), where its inclusion improves the $z\_1$ silhouette score from 0.43 to 0.61.
>
> This bottleneck compels the encoder to route the necessary intra-class variation into the residual subspace $\mathbf{z}\_0$ to satisfy the reconstruction objective, a dynamics formalized in **Lemma 2**. To ensure this residual subspace remains class-agnostic, we apply a targeted adversarial classifier, whose efficacy is formally bounded in **Lemma 1**. Finally, to prevent this pressure from causing a degenerate collapse - a failure mode we demonstrate empirically in **Figure 7** - a counter-balancing mutual information objective ($\mathcal{L}\_{I}$) ensures $\mathbf{z}\_0$ remains informative.
>
> We are confident that these revisions, which directly address the points in **Questions 5 and 6**, have made the paper's principled design significantly more accessible.

---

> ### Author Response · Authors · 2025-12-01
>
> ### Reviewer comment
>
> > "The proposed method introduces too many objectives (at least seven within an adversarial scheme), which overly complicates the framework..."
>
> **Response:** We agree that the framework has several components and appreciate the opportunity to justify its design. The model's structure is the result of a principled approach where each component is introduced to solve a specific, predictable failure mode of a simpler baseline. We will walk through this design process, showing how each objective is a necessary instrument, grounded in our theoretical analysis and validated by our empirical results.
>
> Our model's design follows from addressing the failures of a naive approach to this factorization problem. A simple baseline would be to train an encoder to produce a class-predictive embedding $z\_1$ (via a classification loss $\mathcal{L}\_y$) and a residual $z\_0$, using both for reconstruction. While simple, this approach is susceptible to several critical failure modes, which our additional components are designed to solve not in isolation, but as an interdependent system.
>
> First, there is the problem of **critical information leakage**. Without any explicit constraint, the encoder will readily encode supervised information into $z\_0$. This failure is evident in the t-SNE visualizations in **Figure 4b**, where $z\_0$ clearly retains a class-separated geometry. To counteract this, we introduce a targeted **adversarial classifier on $z\_0$ ($\mathcal{L}\_{G,R}$)**. Its effectiveness is formally analyzed in **Section 4.2**, where we derive **Lemma 1**, providing a quantitative upper bound on the mutual information $I(y; z\_0)$. The necessity of this direct adversarial pressure, even in the presence of other constraints, is highlighted in **Table 1**. A model with the structural loss but lacking the adversary (+ $\mathcal{L}\_y$ + $\mathcal{L}\_\tau$) achieves a probe-gap of only 41.19 pp, demonstrating that structural constraints alone are insufficient to prevent significant leakage.
>
> However, the adversary introduces a new dynamic that can lead to a second failure mode: **concept pollution**. The pressure on $z\_0$ can incentivize the encoder to push all variation - both class-relevant and stylistic - into $z\_1$. To solve this, we introduce the **structural prototype loss ($\mathcal{L}\_\tau$)**. As motivated in **Section 4.3**, this loss enforces extreme intra-class compactness, turning $z\_1$ into a tight **information bottleneck** that is inefficient for carrying non-class details. This mechanism is formalized in **Lemma 2**. The critical interplay between these components is revealed in our ablations. The configuration that includes all other objectives but removes *only* the prototype loss (+ $\mathcal{L}\_y$ + $\mathcal{L}\_{G,R}$ + ($\mathcal{L}\_I$, $\mathcal{L}\_{G,\pi}$)) sees its $z\_1$ silhouette score collapse from 0.40 (full model) to just 0.31, and its $z\_1$ ARI drops from 0.90 to 0.74. This confirms that even with a strong class-agnostic adversary, the concept subspace becomes polluted unless this explicit structural pressure is applied.
>
> Finally, the powerful adversarial pressure on $z\_0$ risks a third failure mode: a **degenerate residual representation**, where the generator learns to simply ignore $z\_0$. To prevent this, we introduce the **InfoGAN-style mutual information objective ($\mathcal{L}\_I$)**. This acts as a crucial counter-pressure, ensuring $z\_0$ remains an informative channel for reconstruction, a principle formally grounded in the Barber-Agakov lower bound (**Appendix A.2.2, Proposition 1**). We provide clear visual evidence of this failure mode and its solution in **Figure 7**, where a model without $\mathcal{L}\_I$ produces a collapsed, useless representation under latent traversal, while the full model learns meaningful, class-agnostic factors.
>
> In conclusion, the complexity of our framework is not incidental but is the result of a sequence of principled solutions to the specific, observable failure modes and their interdependencies. Each component is a necessary instrument, grounded in our theoretical analysis and validated by our empirical results, in the carefully orchestrated system required for robust bipartite factorization.

---

> ### Author Response · Authors · 2025-12-01
>
> ### Reviewer comment
>
> > "A major concern is the lack of comparisons to prior work (e.g.[1],)..."
>
> **Response:** We respectfully wish to clarify that the paper does contain a direct, quantitative comparison to a key prior work, and then explain the principled reason for this choice over other potential baselines like Locatello et al. (2020).
>
> First, we want to point to **Appendix C and Table 4** of our original submission (which has been moved to the main paper as **Table 2** in the revised manuscript), where we conduct a detailed comparison against **Fader Networks**. We chose Fader Networks as our primary quantitative baseline because its problem setting is the most direct and scientifically relevant competitor to ours: both methods leverage a **single, known semantic label** to achieve a form of supervised disentanglement. This allows for a direct comparison on the task of isolating a known concept. Our quantitative results show that PRISM achieves lower label leakage and better reconstruction, demonstrating a superior bipartite factorization.
>
> The important work of Locatello et al. (2020), which the reviewer correctly highlights, addresses a different and complementary form of weak supervision. Their method uses an **unlabeled structural signal** (pairs of images with unknown shared factors) to achieve the goal of **full disentanglement**. In contrast, our method uses a **semantic signal** (a class label) to achieve the more targeted goal of **bipartite factorization**.
>
> Given these distinct problem formulations - differing in both data assumptions (labeled vs. paired data) and scientific goals (semantic purity vs. full structural alignment) - a direct quantitative comparison would not yield an informative measure of relative performance. We apologize that our rationale for focusing on Fader Networks was not made more explicit in the submission, and we hope this explanation clarifies that our choice was a principled one, aimed at ensuring the most scientifically relevant comparison.
>
> ### Reviewer comment
>
> > "...and the absence of evaluation on common disentanglement metrics (e.g., DCI, FactorVAE score, MIG as in), which makes it hard to assess the method’s effectiveness relative to prior work. Instead, the paper reports probe-gap (without reference or justification) and ARI, which appears orthogonal to disentanglement quality."
>
> **Response:** First, to correct a key misunderstanding, we want to clarify that we do evaluate our method using standard, well-established disentanglement metrics. **In Appendix C, Table 4 of the original submission (now presented as Table 2 in the main paper of the revised version), we present a quantitative comparison against the Fader Networks baseline using both the DCI (Disentanglement, Completeness, Informativeness) and SAP (Separated Attribute Predictability) scores.**
>
> Second, we would like to justify our use of the probe-gap and ARI metrics in the main paper's ablation study (Table 1). Our primary research objective is **bipartite factorization**: the supervised partitioning of a latent space into a concept-relevant subspace ($z\_1$) and a class-agnostic residual subspace ($z\_0$). To evaluate this, we require metrics that directly measure the quality of this information partition. A linear probe is a standard and rigorous tool for assessing the degree of linearly separable information already present in a representation [1]. This evaluation protocol is common practice for evaluating learned representations, including within the field of disentanglement [2]. A low linear probe accuracy on $z\_0$ provides a strong, unambiguous signal of successful information purging. The **probe-gap** then directly quantifies the *degree* of separation between the subspaces. Similarly, we use the **Adjusted Rand Index (ARI)** not as a general disentanglement score, but as a diagnostic tool: a high ARI on $z\_1$ validates its class-centric structure, while a low ARI on $z\_0$ confirms the weakness of the class signal, i.e., low information leakage.
>
> In summary, our evaluation strategy was designed to be comprehensive: we use standard metrics like DCI and SAP to verify the disentangled structure of our residual subspace in comparison to baselines, and we employ targeted metrics like the probe-gap and ARI to perform a rigorous, controlled analysis of our primary contribution - bipartite factorization.
>
> ### References
>
> [1] Alain, G. and Bengio, Y., Understanding intermediate layers using linear classifier probes, in ICLR 2017.
>
> [2] Locatello, F. et al., Challenging common assumptions in the unsupervised learning of disentangled representations, in ICML 2019.

---

> ### Author Response · Authors · 2025-12-01
>
> ### Reviewer comment
>
> > "Why is the structural loss (L196) necessary for disentanglement? Isn’t this objective orthogonal to disentanglement of the representation?"
>
> **Response:** The structural loss ($\mathcal{L}\_\tau$) is not intended to disentangle $z\_1$ itself, but rather to act as a crucial pressure that enables the factorization of the *entire latent space*. Its role is to enforce extreme intra-class compactness on the concept-relevant subspace $z\_1$, effectively turning $z\_1$ into a tight **information bottleneck**. By making $z\_1$ an inefficient channel for encoding rich, non-class stylistic details (like stroke thickness or angle), this bottleneck **compels the encoder to route this necessary reconstruction information into the more accommodating residual subspace $z\_0$**.
>
> This emergent routing dynamics is the central principle formalized in **Lemma 2 (Section 4.3)**, which establishes a direct theoretical link between the compactness of $z\_1$ (controlled by $\mathcal{L}\_\tau$) and the amount of information preserved in $z\_0$. Our ablation study in **Table 1** provides strong empirical validation for this. When we include the prototype loss alongside the classifier and adversary (+ $\mathcal{L}\_y$ + $\mathcal{L}\_{G,R}$ + $\mathcal{L}\_\tau$), the $z\_1$ silhouette score (a measure of compactness) jumps from 0.43 to 0.61. This increased pressure on $z\_1$ directly contributes to a cleaner factorization.
>
> Therefore, the structural loss is not orthogonal to our goal; it is a key causal mechanism that creates the necessary pressure on one subspace to structure the information flow into the other, enabling the clean bipartite factorization we aim to achieve.
>
> ### Reviewer comment
>
> > "Why do we have to maximize $I(z\_0; x)$? Doesn’t the reconstruction loss already encourage $z\_0$ to contain information for reconstructing x?"
>
> **Response:** While the reconstruction loss does indeed create a "pull" for information into the entire latent code $z = (z\_1, z\_0)$, this is counteracted by the strong adversarial "push" from $\mathcal{L}\_{G,R}$, which encourages the encoder to make $z\_0$ uninformative about the class label $y$.
> Without a targeted counter-pressure, this adversarial push can lead to a **degenerate solution** where the generator learns to simply ignore $z\_0$ altogether, causing the representation to collapse. The objective to maximize the mutual information $I(z\_0; x)$ (via our tractable InfoGAN-style loss $\mathcal{L}\_I$) provides this necessary counter-pressure. It explicitly encourages $z\_0$ to retain salient, class-agnostic information that is vital for reconstruction.
>
> We provide direct empirical evidence of this failure mode in **Figure 7**. A model trained with the adversary but *without* the $\mathcal{L}\_I$ objective (Figure 7a) learns a collapsed residual representation; traversing the latent space of $z\_0$ produces no meaningful change in the output. In contrast, the full model (Figure 7b), which includes the $\mathcal{L}\_I$ objective, learns to encode meaningful, class-agnostic attributes (like stroke angle and thickness) in $z\_0$, demonstrating that this term is essential for preventing collapse and ensuring a useful, structured residual subspace.
>
> ### Reviewer comment
>
> > "What is $R\_X(d\_{\text{rec}})$ in L313 and how is it measured?"
>
> **Response:** We apologize for not making an explicit link to the explanation of the symbol's meaning, which we provided in the Appendix A. $R\_X(d\_{\text{rec}})$ is the theoretical **Shannon rate-distortion function**. We provide a formal definition for this term and its role in our analysis in **Appendix A.4.1 (lines 768-770 of the original submission)**, citing the foundational work by Berger [1].
>
> Crucially, as we state in the paper (lines 315-317 of the original submission), this function is **intractable to measure directly** in practice. Its purpose in our work is not for direct measurement or optimization, but to serve as a cornerstone for our formal analysis in **Lemma 2**. By linking the required reconstruction fidelity ($d\_{\text{rec}}$) to a minimum required information rate ($R\_X$), it allows us to establish a principled theoretical foundation. From this foundation, we derive clear, falsifiable hypotheses about the architectural levers that control information flow in our model (i.e., reconstruction quality, subspace dimensionality $d\_1$, and intra-class compactness $\delta\_\tau$). We then validate these theoretical predictions with the controlled empirical experiments presented in **Section 5.3 (Figure 2)**.
>
> In short, $R\_X(d\_{\text{rec}})$ is a standard, formal tool used to build our theoretical argument, which in turn guides and justifies our empirical investigation. We revised the text around line 313 to make the pointer to its definition in the appendix more explicit.
> ### Reference
>
> [1] Berger, T. Rate Distortion Theory: A Mathematical Basis for Data Compression. 1971.

---

> ### Author Response · Authors · 2025-12-01
>
> ### Reviewer comment
>
> > "How did the authors select and tune the numerous hyperparameters for the loss-weighting coefficients?"
>
> **Response:** The final hyperparameters for our full PRISM model are listed in **Appendix B.3, Table 3**. For the main ablation study presented in **Section 5.2 (Table 1)**, our goal was to perform a fair and controlled comparison of how different combinations of our loss components affect the model's ability to perform bipartite factorization. To achieve this, as described in lines 333-338 of the original submission, each model configuration (e.g., "Baseline + $\mathcal{L}\_y$", "Baseline + $\mathcal{L}\_{G,R}$", etc.) was allocated an identical, fixed hyperparameter search budget. For each configuration, we performed a random search over the loss-weighting coefficients and selected the set of hyperparameters that yielded the best performance on our primary evaluation metric for factorization quality: the **probe-gap**. This protocol ensures that the superior performance of the full model is a result of its more comprehensive architecture, not a greater tuning effort.
>
> To further ensure full reproducibility of our results, we have included the complete source code for our experiments in the supplementary material and intend to release it publicly upon publication.
>
> ### Reviewer comment
>
> > "Please add equation numbering."
>
> > "Please consider redrawing Figure 1 for readability; the text in the figure is too small to read and the figure looks too complex."
>
> **Response:** We thank the reviewer for these concrete and helpful suggestions to improve the paper's presentation.
>
> You are absolutely right to point out the lack of equation numbers; we apologize for this significant oversight. We added them throughout the paper in the revised version to improve clarity and ease of reference.
>
> We also agree that Figure 1, in its current form, is too dense and the text is difficult to read. We revised the figure using a larger font size to ensure all components and labels are clearly legible in the final manuscript.

---

### Official Review · Reviewer_Wv49 · 2025-11-01

**Soundness:** 2
**Presentation:** 2
**Contribution:** 2
**Rating:** 6
**Confidence:** 2

**Summary:**

This paper addresses the problem of bipartite factorization in generative models—separating a supervised semantic factor (e.g., class label) from all remaining variation. The proposed model, PRISM, uses a combination of adversarial training, prototype-based supervision, and information bottlenecks to route intra-class variation into a residual latent subspace. The paper also provides an information-theoretic analysis explaining the conditions under which such routing occurs, and validates these predictions with targeted experiments on Morpho-MNIST and qualitative disentanglement scenarios on CelebA.

**Strengths:**

The paper presents a well-motivated and conceptually clean solution to the challenging problem of bipartite latent space factorization. By explicitly separating a supervised semantic factor from residual variation, PRISM provides a principled alternative to fully unsupervised disentanglement, which is known to be theoretically ill-posed. The architecture combines prototype-based regularization, adversarial class removal, and mutual information objectives, producing a latent structure that aligns tightly with the intended factorization. This combination is both innovative and well-justified, and the design decisions are guided by a formal information-theoretic framework.

The experimental validation is strong and carefully targeted. The ablation studies and controlled manipulations of model capacity, concept subspace dimensionality, and structural regularization provide direct empirical support for the theoretical predictions (Lemmas 1 and 2). The qualitative results on CelebA, including attribute swapping and latent traversals, further demonstrate the model’s ability to cleanly isolate concept-relevant features from residual variation. Overall, the work provides a compelling combination of theory, architectural design, and empirical validation that advances the field of structured representation learning.

**Weaknesses:**

Despite its strengths, the proposed approach is fairly complex, involving multiple adversarial networks, prototype updates, and mutual information estimators. This could pose challenges for reproducibility and practical adoption, especially for users without careful hyperparameter tuning or computational resources. Furthermore, while the theoretical analysis provides upper and lower bounds on information leakage and routing, it relies on strong assumptions about equilibrium convergence in multi-agent optimization, which may not hold in practice and could limit the applicability of the formal guarantees.

The empirical evaluation, though thorough in testing theoretical predictions, is somewhat limited in scope. The primary quantitative evaluations are conducted on synthetic or semi-synthetic datasets (Morpho-MNIST), with real-world datasets (CelebA) evaluated mostly qualitatively. Additionally, while comparisons are made to Fader Networks, the study does not thoroughly benchmark against other recent methods for supervised factorization or disentanglement. Broader quantitative evaluation on natural datasets and more diverse baselines would strengthen the evidence for PRISM’s practical effectiveness.

**Questions:**

- How sensitive is PRISM to the choice of hyperparameters and the stability of the adversarial training? Could slight variations in weights or network capacity lead to a collapse of the bipartite factorization, and how might this affect practical deployment?

- Can the proposed framework generalize to multi-factor supervised settings, where multiple attributes need to be separated simultaneously, and if so, what modifications to the architecture or objectives would be necessary?

---

> ### Author Response · Authors · 2025-12-01
>
> We sincerely thank you for your time and for providing a detailed and constructive review. We appreciate your assessment of our work as a "well-motivated and conceptually clean solution" and your recognition of the "strong and carefully targeted" experimental validation. We are glad you found the combination of theory and architectural design compelling. Below, we address your concerns regarding complexity, theoretical assumptions, and evaluation scope. We also provide answers to your questions regarding hyperparameter sensitivity, stability, and the framework's potential for multi-factor generalization. We believe these clarifications effectively address your concerns.
>
> ### Reviewer comment
>
> > "Despite its strengths, the proposed approach is fairly complex, involving multiple adversarial networks, prototype updates, and mutual information estimators. This could pose challenges for reproducibility and practical adoption, especially for users without careful hyperparameter tuning or computational resources."
>
> **Response:** We thank the reviewer for this crucial and practical point. We agree that the framework is multifaceted, and we appreciate the opportunity to justify its design, not only in terms of its theoretical underpinnings but also its practical viability. The model's structure is the result of a principled approach where each component is introduced to solve a specific, predictable failure mode of a simpler baseline.
>
> Our model's design follows from addressing the failures of a naive approach. First, there is the problem of **critical information leakage**: without any explicit constraint, an encoder will readily encode supervised information into the residual subspace $\mathbf{z}\_0$. This failure is evident in the t-SNE visualizations in **Figure 4b**, where $\mathbf{z}\_0$ clearly retains a class-separated geometry. To counteract this, we introduce a targeted **adversarial classifier on $\mathbf{z}\_0$ ($\mathcal{L}\_{G,R}$)**. Its effectiveness is formally analyzed in **Section 4.2**, where we derive **Lemma 1**, providing a quantitative upper bound on the mutual information $I(y; \mathbf{z}\_0)$.
>
> However, the adversary introduces a new dynamics that can lead to a second failure mode: **concept pollution**. The pressure on $\mathbf{z}\_0$ can incentivize the encoder to push all variation - both class-relevant and stylistic - into $\mathbf{z}\_1$. To solve this, we introduce the **structural prototype loss ($\mathcal{L}\_\tau$)**. As motivated in **Section 4.3**, this loss enforces extreme intra-class compactness, turning $\mathbf{z}\_1$ into a tight **information bottleneck** that is inefficient for carrying non-class details, a mechanism formalized in **Lemma 2**. The critical interplay between these components is revealed in our ablations: the configuration in **Table 1** that removes *only* the prototype loss sees its $\mathbf{z}\_1$ silhouette score collapse from 0.40 (full model) to just 0.31, confirming that the concept subspace becomes polluted without this explicit structural pressure.
>
> Finally, the powerful adversarial pressure on $\mathbf{z}\_0$ risks a third failure mode: a **degenerate residual representation**, where the generator learns to simply ignore $\mathbf{z}\_0$. To prevent this, we introduce the **InfoGAN-style mutual information objective ($\mathcal{L}\_I$)**. This acts as a crucial counter-pressure, ensuring $\mathbf{z}\_0$ remains an informative channel for reconstruction, a principle formally grounded in the Barber--Agakov lower bound (**Appendix A.2.2, Proposition 1**). We provide clear visual evidence of this failure mode and its solution in **Figure 7**, where a model without $\mathcal{L}\_I$ produces a collapsed, useless representation under latent traversal, while the full model learns meaningful, class-agnostic factors.
>
> Crucially, this principled design directly addresses the valid concerns about **hyperparameter tuning and computational resources**. Regarding resources, the *additional* auxiliary networks (latent classifiers) are lightweight MLPs, while the image discriminator is standard for any GAN baseline; thus, the marginal computational overhead over methods like Fader Networks is negligible.
> To ensure fairness and reproducibility, we allocated an identical search budget to each configuration in our ablation study (**Section 5.2**). The final, robust hyperparameters are listed in **Appendix B, Table 3**; the full source code is provided in the supplementary material and will be released publicly upon publication.

---

> ### Author Response · Authors · 2025-12-01
>
> ### Reviewer comment
>
> > "...while the theoretical analysis provides upper and lower bounds on information leakage and routing, it relies on strong assumptions about equilibrium convergence in multi-agent optimization, which may not hold in practice and could limit the applicability of the formal guarantees."
>
> **Response:** We thank the reviewer for highlighting this crucial point, and we are in complete agreement. The assumption of convergence to a stable equilibrium in a complex adversarial game is indeed a strong one, and we were careful to state this explicitly as a precondition for our analysis at the beginning of **Section 4.1** (lines 240-243 of the original manuscript).
>
> To address this reality, and bridge the gap between the theory and practice, we provided a **Finite-Error Analysis in Appendix A.2**. Rather than relying solely on zero-error convergence, we derived our bounds based on *bounded error terms* ($\epsilon\_D, \eta$). This formulation ensures that our formal guarantees do not collapse if the model fails to reach a perfect equilibrium; instead, they degrade gracefully. As long as the adversarial errors are bounded (i.e., the discriminator is "good enough"), the upper bound on information leakage remains valid.
>
> Furthermore, the primary utility of our theory is to serve as a **prescriptive design guide**. As demonstrated in **Section 5.3 and Figure 2**, the causal relationships predicted by our analysis (specifically Lemma 2) hold empirically. We show that manipulating the theoretical levers - reconstruction fidelity and structural regularization - produces precisely the predicted shifts in information routing. This confirms that our theoretical model accurately describes the system's dynamics and provides a powerful tool for controlling the factorization, even in the absence of perfect equilibrium.

---

> ### Author Response · Authors · 2025-12-01
>
> ### Reviewer comment
>
> > "The empirical evaluation, though thorough in testing theoretical predictions, is somewhat limited in scope. The primary quantitative evaluations are conducted on synthetic or semi-synthetic datasets (Morpho-MNIST), with real-world datasets (CelebA) evaluated mostly qualitatively. Additionally, while comparisons are made to Fader Networks, the study does not thoroughly benchmark against other recent methods for supervised factorization or disentanglement. Broader quantitative evaluation on natural datasets and more diverse baselines would strengthen the evidence for PRISM’s practical effectiveness."
>
> **Response:** We thank the reviewer for this thoughtful critique. We fully appreciate the preference for evaluation on complex, natural datasets and for comparison against a broad range of modern methods. However, our experimental design was constrained by two fundamental realities: the lack of complete ground-truth factors in real-world data and the incompatibility of supervision signals across recent methods.
>
> First, regarding our reliance on Morpho-MNIST for quantitative metrics, we chose this dataset to ensure rigorous measurement. Standard disentanglement metrics, such as DCI and SAP, assume access to a complete set of independent generative factors. In Morpho-MNIST, these factors are known, independent, and controllable, allowing us to mathematically verify our theoretical claims (e.g., Lemma 2) and to quantify exactly how much information leaks between subspaces.
>
> In contrast, although CelebA provides 40 binary attributes, these cannot serve as a valid ground-truth set for rigorous disentanglement benchmarking for two reasons:
>
> 1. **Incompleteness:** The attributes do not capture important residual factors such as *head pose, lighting direction,* and *background texture*. Because our residual subspace $\mathbf{z}\_0$ must encode these factors to satisfy the reconstruction objective, calculating metrics like DCI using only the 40 labeled attributes would yield misleading informativeness scores: the subspace would appear to contain "unknown" noise that is actually salient geometric information.
> 2. **Correlations:** Metrics like DCI assume that generative factors are independent. CelebA attributes are highly correlated (e.g., "Male" and "Mustache"). This violates the metric's assumptions and makes it difficult to distinguish true leakage from learned correlations.
>
> For these reasons we used CelebA primarily for qualitative evaluation (**Figure 3**) and for attribute swapping, which provide a practical proxy for factorization quality when reliable ground-truth metrics are unavailable.
>
> Second, regarding comparisons to “other recent methods,” we emphasize that the literature on disentanglement spans several distinct supervision regimes and problem formulations. A significant body of recent work (e.g., [1], [2]) focuses on weakly supervised settings that require paired data or group perturbations. Comparing PRISM (which operates in the standard supervised setting) to those methods would be an apples-to-oranges comparison because the data requirements differ.
>
> Within the supervised setting we selected Fader Networks [3] as our primary baseline because it is the most direct architectural predecessor to our approach. Like PRISM, Fader Networks relies on adversarial training to remove a specific attribute. By benchmarking against it (**Appendix C, Table 4** of the original submission, now in **Section 5.4, Table 2**) we hold the learning mechanism (adversarial training) constant and thereby isolate the contribution of our bipartite factorization strategy relative to the standard invariance strategy. We have expanded **Section 5.4** to make these distinctions explicit.
>
> ### References
>
> [1] Locatello, F., et al. Weakly-supervised disentanglement without compromises. *ICML*, 2020.
>
> [2] Locatello, F., et al. Disentangling factors of variation using few labels. *ICLR*, 2020.
>
> [3] Lample, G., et al. Fader networks: Manipulating images by sliding attributes. *NeurIPS*, 2017.

---

> ### Author Response · Authors · 2025-12-01
>
> ### Reviewer comment
>
> > "How sensitive is PRISM to the choice of hyperparameters and the stability of the adversarial training? Could slight variations in weights or network capacity lead to a collapse of the bipartite factorization, and how might this affect practical deployment?"
>
> **Response:** We thank the reviewer for this important practical question. Our framework's stability comes from its principled design, where sensitivities are both understood and, in many cases, predicted by our theoretical analysis.
>
> Regarding hyperparameter sensitivity, our ablation study in **Table 1** shows that while the relative weighting of loss terms is indeed important for achieving the optimal partition, the full model is robust. The low standard deviation reported across five independent seeds for all key metrics indicates that the training is stable and the results are reliable.
>
> The reviewer astutely asks about conditions that could lead to a collapse of the factorization, and we would like to highlight one such scenario that is both predicted by our theory and was observed in practice. Our theoretical analysis in **Lemma 2** provides a direct answer: the amount of information routed into the residual subspace $z\_0$ is lower-bounded by the total information demand required for reconstruction, represented by the rate-distortion function $R\_X(d\_{\text{rec}})$. This creates a "pull" for information that is directly proportional to the reconstruction fidelity. If the reconstruction quality is poor, this pull is weak, and the factorization will indeed fail to materialize, as there is insufficient pressure to route variation into $z\_0$.
>
> This is not a weakness of the framework, but rather an empirical confirmation of its core theoretical principle. We observed this exact behavior during our experiments on the high-resolution CelebA dataset. An initial model using a simple Mean Squared Error loss did not achieve sufficient reconstruction fidelity, which hindered the clean separation of factors. To increase the information demand, we switched to a stronger VGG-based perceptual loss. This improved reconstruction quality, strengthened the information "pull," and in turn enabled the clean, effective bipartite factorization shown in **Figure 3**.
>
> Finally, to prevent the other primary mode of collapse - where the generator learns to simply ignore $z\_0$ - our framework explicitly includes the InfoGAN-style objective ($\mathcal{L}\_I$), which provides a crucial counter-pressure to ensure the residual subspace remains informative, a failure mode we demonstrate in **Figure 7**. Thus, the framework's key sensitivities are well-understood and directly addressed by its design.
>
> ### Reviewer comment
>
> > "Can the proposed framework generalize to multi-factor supervised settings, where multiple attributes need to be separated simultaneously, and if so, what modifications to the architecture or objectives would be necessary?"
>
> **Response:** We thank the reviewer for this insightful question about the framework's extensibility. The core principles of PRISM are conceptually modular and could be generalized to a multi-factor setting, but we also wish to be transparent about the significant practical challenges such an extension would entail.
>
> The conceptual extension would involve moving from a bipartite partition $(z\_1, z\_0)$ to a multipartite structure $(z\_1, z\_2, \dots, z\_K, z\_0)$. Architecturally, this would require a dedicated concept subspace $z\_k$, classifier $C\_k$, and prototype loss $\mathcal{L}\_{\tau,k}$ for each of the $K$ supervised attributes. The latent-residual discriminator would then be extended into a multi-task adversary with $K$ prediction heads, tasked with purging information about all $K$ attributes from the shared residual subspace $z\_0$.
>
> However, this approach introduces two major practical hurdles. First, it would naturally require access to datasets with rich, multi-attribute labels for every factor one wishes to disentangle. Second, and more critically, the optimization dynamics would become substantially more complex. As the reviewer astutely noted in the earlier comments, our current bipartite model already involves a careful balance of competing objectives. In a multi-factor setting, balancing the adversarial and structural pressures across $K$ distinct concept subspaces and a single residual subspace would present a formidable hyperparameter tuning challenge, making the system significantly harder to stabilize.
>
> While these challenges are non-trivial and would require significant further research, we believe that the core mechanisms of PRISM - creating targeted information bottlenecks for supervised factors while applying targeted adversarial pressure to purify a shared residual space - provide a robust and principled template for future work exploring these more complex factorization problems.

---

### Meta-Review · Area_Chair_wm6J · 2026-01-04

**Summary:**

This work proposes a representation framework that aims to disentangle labeled semantics from all other unlabeled ones. The proposed technique is an adversarial mechanism that encourages the part of the latent components to be class-relevant and the others to be class independent. Information-theoretic bounds were characterized and the method was tested over datasets such as Morpho-MNIST.

Strength:
The reviewers acknowledged that the motivation of the paper is reasonable and the proposed approach is conceptually clean. Multiple reviewers noticed that the work has information-theoretic bounds on information leakage, offering in-depth understanding to the problem under consideration.

Weaknesses:

However, the reviewers also pointed out that the organization and clarity might be a concern -- it is hard to clearly follow the technical development and theoretical claims. More importantly, the proposed method is too complex, involving multiple adversarial networks, prototype updates, and mutual information estimators. This raises concerns in hyperparameter tuning and sensitivity issues.
A reviewer also mentioned that the assumptions of the theory may not be practical.
In terms of experiments, there is a lack of comparison with general disentanglement baselines. Commonly used disentanglement metrics were also not considered. The data like (Morpho-MNIST) might be too simple.


Summary: The work starts with an interesting motivation. However, the proposed approach does seem to have an overly complex loss. The conceptual network has 7 losses added together. The adversarial loss has 5 terms that need to be balanced. The lack of succinctness is a major concern and the AC concurs the comments on hyperparameter tuning challenges. The quantitative experiments can also be enhanced to use more real world data.

**Reviewer Concerns:**

The rebuttal explained the necessity of the complex loss function yet did not alleviate the concern of difficulties in tuning and parameter sensitivity.

The authors revised their theorem to allow some finite errors.

The authors also explained that they did use disentanglement baselines and well established metrics. But they also acknowledged that more complex data is only used for qualitative evaluation.

**Reviewer Scores:**

Reviewer Wv49 (rating 6) found that the loss is too complex, that the theorems rely on strong assumptions, that the empirical result is limited, and that no recent baselines are considered.  The rebuttal explained the rationale behind the loss, the theory and why the empirical results are designed this way.

Reviewer 4kLh (rating 2) also complained about too many objectives, hardness of tuning hyperparameters, lack of comparisons, and that the data used being too simple. The rebuttal objected to some of the complaints.

Reviewer MCZj (rating 6) found the paper intuitive, but math did not add much. The rebuttal provided discussions.

The concerns of Wv49 and 4kLh are valid. These concerns are not easy to be alleviated by ablation study or arguing that the many losses are proposed following principled derivation. The result of such a loss is indeed hard to balance and hard to pick hyperparameters. The reviewers may not be convinced to increase the scores.

---

### Decision · Program_Chairs · 2026-01-26

Reject